# Macrophages promote endothelial-to-mesenchymal transition via MT1-MMP/TGFβ1 after myocardial infarction

Laura Alonso-Herranz[1], Álvaro Sahún-Español[2], Ana Paredes[1], Pilar Gonzalo[2], Polyxeni Gkontra[2], Vanessa Núñez[1], Cristina Clemente[2,3], Marta Cedenilla[1], María Villalba-Orero[1], Javier Inserte[4,5], David García-Dorado[4,51], Alicia G Arroyo[2,3]*, Mercedes Ricote[1]*

[1]Myocardial Pathophysiology Area, Centro Nacional de Investigaciones Cardiovasculares (CNIC), Madrid, Spain; [2]Vascular Pathophysiology Area, Centro Nacional de Investigaciones Cardiovasculares (CNIC), Madrid, Spain; [3]Molecular Biomedicine Department, Centro de Investigaciones Biológicas Margarita Salas (CIB-CSIC), Madrid, Spain; [4]Cardiovascular Diseases Research Group, Vall d'Hebron University Hospital and Research Institute (VHIR), Barcelona, Spain; [5]CIBER de Enfermedades Cardiovasculares (CIBERCV), Madrid, Spain

*For correspondence:
agarroyo@cib.csic.es (AGA);
mricote@cnic.es (MR)

[†]Deceased

Competing interests: The authors declare that no competing interests exist.

**Abstract** Macrophages (Mφs) produce factors that participate in cardiac repair and remodeling after myocardial infarction (MI); however, how these factors crosstalk with other cell types mediating repair is not fully understood. Here we demonstrated that cardiac Mφs increased the expression of *Mmp14* (MT1-MMP) 7 days post-MI. We selectively inactivated the *Mmp14* gene in Mφs using a genetic strategy (*Mmp14*^f/f:*Lyz2*-Cre). This conditional KO (MAC-Mmp14 KO) resulted in attenuated post-MI cardiac dysfunction, reduced fibrosis, and preserved cardiac capillary network. Mechanistically, we showed that MT1-MMP activates latent TGFβ1 in Mφs, leading to paracrine SMAD2-mediated signaling in endothelial cells (ECs) and endothelial-to-mesenchymal transition (EndMT). Post-MI MAC-Mmp14 KO hearts contained fewer cells undergoing EndMT than their wild-type counterparts, and *Mmp14*-deficient Mφs showed a reduced ability to induce EndMT in co-cultures with ECs. Our results indicate the contribution of EndMT to cardiac fibrosis and adverse remodeling post-MI and identify Mφ MT1-MMP as a key regulator of this process.

## Introduction

Although the mortality rate from cardiovascular disease (CVD) has declined over the last 50 years, myocardial infarction (MI) continues to be one of the most lethal diseases worldwide, and many pathologies arise from adverse remodeling after cardiac injury (e.g. heart failure and cardiac rupture) (*Benjamin et al., 2017*). Therefore, new strategies are urgently needed to improve cardiac repair and function after MI. MI is usually provoked by plaque rupture in a coronary artery, resulting in insufficient oxygen supply to the myocardium, which undergoes necrosis. The onset of MI triggers a cascade of events, including cardiomyocyte (CM) death, acute inflammation, angiogenesis, and scar formation (*Frangogiannis, 2015*). Cardiac healing is impaired by both excessive and insufficient expansion of macrophages (Mφs), prompting interest in the potential of Mφs as therapeutic targets for CVD (*Dick et al., 2019*; *Panizzi et al., 2010*; *van Amerongen et al., 2007*). Nonetheless, the exact Mφ phenotypes and mechanisms that might enhance tissue repair are not clearly defined.

We and others have shown that post-injury Mφs are distinct from resident cardiac Mφs (*Bajpai et al., 2019*; *Dick et al., 2019*; *Walter et al., 2018*). Monocytes and Mφs are sequentially mobilized from bone marrow (BM) and spleen to the infarcted myocardium (*Nahrendorf et al.,*

2007; Swirski et al., 2009). During day 1 to day 4 after injury, Ly6C[high] inflammatory Mφs clear necrotic cellular debris and damaged extracellular matrix (ECM) from the tissue and attract other immune cells through the secretion of pro-inflammatory cytokines (TNFα, IL1β, and IL6) that further fuel inflammation. Over the course of several days, the inflammatory phase gives way to a healing phase, dominated by the second wave of Mφs, this time Ly6C[low] reparative Mφs with the capacity to dampen inflammation, promote ECM reconstruction, and angiogenesis. These latter Mφs are characterized by the secretion of anti-inflammatory (IL10), angiogenic (VEGF), and pro-fibrotic (TGFβ1) factors, as well as matrix metalloproteinases (MMPs), which promote tissue remodeling (Nahrendorf et al., 2007; Walter et al., 2018). However, how these factors produced by Mφs cross-talk with other cell types and orchestrate cardiac repair is still not fully elucidated.

MMPs are a family of zinc-dependent endopeptidases that have been traditionally associated with the degradation and turnover of ECM components. MMPs are now known to, directly and indirectly, regulate cell behavior and microenvironment through the proteolytic processing of a large variety of molecules, such as membrane receptors and growth factors (Page-McCaw et al., 2007). Membrane type 1 matrix metalloproteinase (MT1-MMP/Mmp14) was the first membrane-anchored MMP to be described (Sato et al., 1994), and the cell surface location provides this enzyme with exclusive functions affecting cellular behavior. MT1-MMP is involved in the degradation of a spectrum of structural matrix proteins (including collagens I, II, and III, fibronectin, and laminin), the proteolytic processing of growth factors and cytokines (e.g. TGFβ and SDF-1), and the activation of other MMPs (e.g. pro-MMP2), expanding MT1-MMP functional pleiotropism.

Previous studies pointed out a deleterious role of MT1-MMP in post-MI cardiac remodeling, mostly related to its collagenase activity in fibroblasts (FBs) (Koenig et al., 2012). Nevertheless, specific Mφ MT1-MMP contribution to cardiac healing response has never been addressed. In this study, we demonstrate that Mφ-restricted Mmp14 inactivation attenuates post-MI left ventricular (LV) dysfunction by preventing endothelial-to-mesenchymal transition (EndMT), maybe accompanied by other concomitant processes, and propose new treatment options for cardiac ischemic disease based on the modulation of Mφ MT1-MMP activity.

## Results

### MI induced the expression of *Mmp14* in Mφs

To gain insight into the potential contribution of Mφ-produced MT1-MMP to cardiac healing, we induced MI in adult mice by permanent coronary ligation (LAD-ligation). Using an established gating strategy (Walter et al., 2018), we isolated cardiac Mφs at 0, 3, 7, and 28 days post-MI (Figure 1—figure supplement 1A–B) and assessed gene expression related to ECM remodeling (Figure 1A). MI induced expression of *Mmp14* (MT1-MMP) and its substrates *Mmp2* and *Col1a1* in Mφs in the heart, reaching maximum levels of expression on day 7 post-MI. In contrast, other MMP family members (*Mmp9* and *Mmp13*) were downregulated after infarction.

### Mφ-restricted MT1-MMP deficiency attenuates LV dysfunction and dilation and reduces collagen deposition after MI

To explore the role of Mφ-derived MT1-MMP in post-MI LV function and remodeling, we generated a Mφ-specific KO mouse for *Mmp14*. *Lyz2*-Cre mice (Clausen et al., 1999) were crossed with *Mmp14*[f/f] mice (Gutiérrez-Fernández et al., 2015), resulting in the deletion of exons 4 and 5 in the floxed *Mmp14* allele in MAC-Mmp14 KO mice (Cre[+] mice) (Figure 1—figure supplement 2A). *Mmp14*[f/f] littermates (Cre[-]) were used as WT controls. No significant differences in either baseline cardiac function (Table 1) or in circulating monocytes and neutrophils were found between genotypes (Table 2, and Figure 1—figure supplement 2B–C). *Mmp14* was efficiently inactivated in BM-derived Mφs (BMDMs) and in cardiac Mφs sorted from 7-day-post-MI MAC-Mmp14 KO hearts; in contrast, *Mmp14* expression did not differ between endothelial cells (ECs) purified from WT and MAC-Mmp14 KO 7-day-post-MI hearts (Figure 1—figure supplement 2D). The low-efficiency recombination of the *Lyz2*-Cre system in monocytes and dendritic cells (Abram et al., 2014) and the lack of *Mmp14* expression in neutrophils (Daseke et al., 2019) make this MAC-Mmp14 KO mouse a suitable model to study Mφ-MT1-MMP role in post-MI LV remodeling and function.

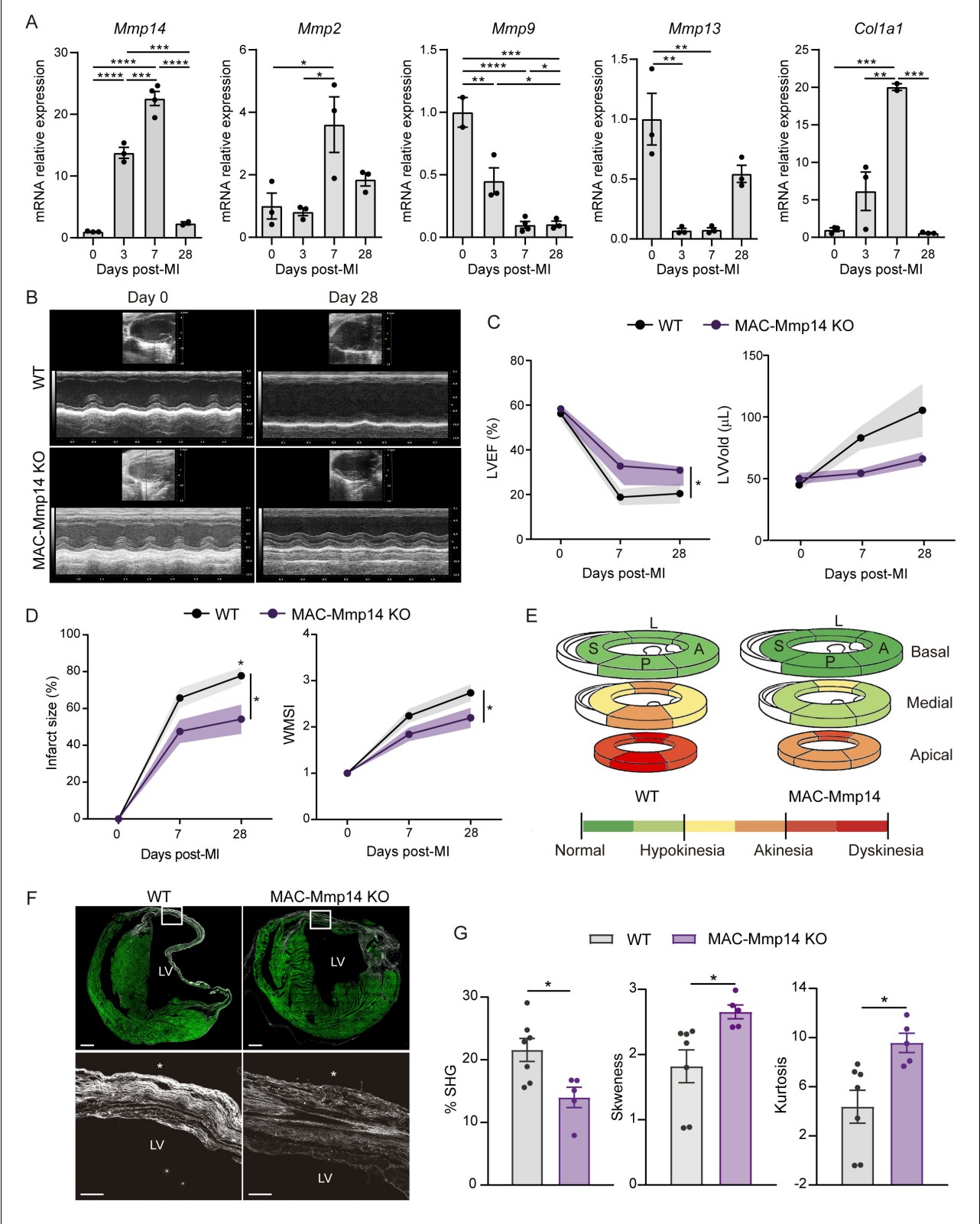

**Figure 1.** Mφ-restricted MT1-MMP deficiency attenuates LV dysfunction and dilation and reduces collagen deposition after MI. (**A**) mRNA expression levels of genes related to ECM-remodeling assessed by qPCR in sorted Mφs at the indicated post-MI stages. Data are means ± SEM of three independent pools of 3–5 mice per time point. One-way ANOVA followed by Tukey's multiple comparisons test. (**B**) Representative LV M-mode long-axis echocardiography views at end-diastole on day 0 and day 28 post-MI in WT and MAC-Mmp14 KO mice. (**C, D**) Post-MI progression of LVEF and

*Figure 1 continued on next page*

*Figure 1 continued*

LVVold (C) and infarct size (percentage of LV with contractility alterations) and WMSI (D) assessed by echocardiography. Data are means ± SEM of 9–10 mice per genotype. Two-way ANOVA followed by Tukey's multiple comparisons test. (E) Quantitative assessment of LV contractility at 28 days post-MI, showing mean scores for every LV segment at the basal, medial, and apical levels throughout all samples. L, lateral; A, anterior; P, posterior; S, septal. Segment scores are colored-coded from green to red: green = normal, yellow = hypokinesia, orange = akinesia, and red = dyskinesia or aneurysm. (F) Representative SHG (white) and MPEF (green) microscopy images of transverse cardiac sections at 28 days post-MI. Scale bar, 500 µm. Magnified views of boxed areas within the infarct are shown in the lower panels. Scale bar, 100 µm. Asterisks mark the epicardium. (G) Percentage SHG, skewness, and kurtosis in infarcts at 28 days post-MI. Data are means ± SEM of 5–7 mice per genotype. Unpaired *t*-test.

The online version of this article includes the following source data and figure supplement(s) for figure 1:

**Source data 1.** Mφ-restricted MT1-MMP deficiency attenuates LV dysfunction and dilation and reduces collagen deposition after MI.

**Figure supplement 1.** Isolation of cardiac Mφs after MI.

**Figure supplement 2.** Mouse model of Mφ-specific MT1-MMP inactivation.

**Figure supplement 2—source data 1.** Mouse model of Mφ-specific MT1-MMP inactivation.

**Figure supplement 3.** The inactivation of Mφ MT1-MMP ameliorates cardiac dysfunction and reduces collagen deposition in a model of transient ischemia.

**Figure supplement 3—source data 1.** The inactivation of Mφ MT1-MMP ameliorates cardiac dysfunction and reduces collagen deposition in a model of transient ischemia.

**Figure supplement 4.** MT1-MMP inactivation does not alter the Mφ phenotype following MI.

**Figure supplement 4—source data 1.** MT1-MMP inactivation does not alter the Mφ phenotype following MI.

Echocardiography revealed that Mφ-inactivation of *Mmp14* ameliorated LV dysfunction and prevented LV dilation post-MI, with MAC-Mmp14 KO mice having a significantly higher LV ejection fraction (LVEF) and lower LV end-diastolic volume (LVVold) (*Figure 1B–C*, and *Video 1*). Infarct size and LV wall motion score index (WMSI) were calculated to assess global and regional cardiac contractility abnormalities by echocardiography (see Materials and methods). This analysis showed that LAD-ligation produced smaller infarcts and a lower WMSI in MAC-Mmp14 KO mice (*Figure 1D–E*), indicating better preservation of cardiac function and less pronounced wall-motion abnormalities in the LV when Mφ MT1-MMP was absent.

MT1-MMP can process a variety of ECM components, so we next analyzed the fibrotic response to MI. Transverse sections of 28-day-post-MI hearts were assessed using a multi-photon laser scanning microscope to capture multi-photon autofluorescence (MPEF) and second harmonic generation (SHG) signals, allowing simultaneous visualization of myocyte components and fibrillary collagen,

**Table 1.** Mφ MT1-MMP inactivation does not affect homeostatic cardiac function.

Echocardiography and electrocardiography comparisons between 10-week-old WT and MAC-Mmp14 KO mice. Data are means ± SEM of 10 mice per group. Unpaired *t*-test. BW, body weight; FS, fraction shortening; HR, heart rate; HW, heart weight; LVEF, LV ejection fraction; LVIDd, LV end-diastolic internal diameter; LVIDs, LV end-systolic internal diameter; LVVold, LV end-diastolic volume; LVVols, LV end-systolic volume.

|  | WT | MAC-Mmp14 KO |
|---|---|---|
| BW (g) | 18.62 ± 2.00 | 19.53 ± 2.25 |
| HW/BW (mg/g) | 4.91 ± 0.28 | 4.94 ± 0.19 |
| HR (beats/min) | 462 ± 14 | 463 ± 17 |
| PR (ms) | 39.82 ± 1.18 | 37.70 ± 1.21 |
| QRS (ms) | 25.41 ± 1.06 | 26.18 ± 0.88 |
| LVEF (%) | 53.52 ± 1.72 | 53.14 ± 2.31 |
| LVVols (µL) | 21.53 ± 1.36 | 19.98 ± 1.60 |
| LVVold (µL) | 25.26 ± 0.96 | 22.22 ± 0.90 |
| FS (%) | 26.58 ± 0.89 | 26.33 ± 1.51 |
| LVIDs (mm) | 3.67 ± 0.06 | 3.54 ± 0.08 |
| LVIDd (mm) | 2.7 ± 0.06 | 2.62 ± 0.10 |

**Table 2.** Mφ MT1-MMP inactivation does not affect circulating bone marrow-derived populations. Hematograms from 10-week-old WT and MAC-Mmp14 KO mice. Data are means ± SEM of eight mice per group.

| | WT | | MAC-Mmp14 KO | |
| --- | --- | --- | --- | --- |
| | % | Cells ($\times 10^3$)/mL | % | Cells ($\times 10^3$)/mL |
| Neutrophils | 9.50 ± 0.98 | 0.74 ± 0.11 | 11.01 ± 1.43 | 0.88 ± 0.13 |
| Lymphocytes | 86.75 ± 1.29 | 6.97 ± 1.00 | 84.79 ± 1.68 | 6.84 ± 0.54 |
| Monocytes | 0.89 ± 0.14 | 0.08 ± 0.02 | 1.20 ± 0.13 | 0.10 ± 0.01 |
| Eosinophils | 2.45 ± 0.44 | 0.21 ± 0.06 | 2.48 ± 0.51 | 0.20 ± 0.04 |
| Basophils | 0.41 ± 0.08 | 0.03 ± 0.01 | 0.53 ± 0.09 | 0.05 ± 0.01 |

respectively. SHG images of the infarct zone (IZ) revealed a highly directional and organized collagen fibril morphology in the WT infarcted group, whereas a less-organized and sparse collagen structure was observed in MAC-Mmp14 KO infarcted hearts (*Figure 1F*). Quantification of SHG signals revealed a marked drop in fibrillary collagen density in MAC-Mmp14 KO infarcted hearts (*Figure 1G*). In addition, the analysis of first-order features of collagen fibrils in SHG images showed augmented skewness and kurtosis in MAC-Mmp14 KO mice, indicating thinner and underdeveloped, disarrayed collagen fibers and thus lower tissue stiffness (*Figure 1G*).

Next, we investigated the effect of Mφ-specific *Mmp14* inactivation during the transient ischemia/reperfusion (I/R) model of MI. Similar to the permanent occlusion model, transient ischemia impaired LV function in both genotypes, but MAC-Mmp14 KO mice had a significantly higher LVEF and lower LV end-systolic and end-diastolic internal diameters (LVIDs and LVIDd, respectively), indicating better preserved LV function and prevention of LV dilation (*Figure 1—figure supplement 3A*). Moreover, SHG imaging analysis after I/R revealed a weaker fibrotic response, with thinner and sparser collagen fibers in MAC-Mmp14 KO hearts (*Figure 1—figure supplement 3B–C*).

These data indicate that Mφ *Mmp14*-deficiency decreases collagen fiber deposition, producing a smaller and underdeveloped fibrotic scar that results in relatively low LV dilation and ameliorated systolic dysfunction after transient or permanent cardiac ischemia.

## *Mmp14* inactivation in Mφs does not influence their phenotype

We first investigated whether *Mmp14* inactivation influenced the Mφ activation program. We assessed classical anti-inflammatory genes expressed by cardiac Mφs at day 7 post-MI (*Walter et al., 2018*) and no differences were found between phenotypes (*Figure 1—figure supplement 4A*). In line with a previous study (*Walter et al., 2018*), most classical pro-inflammatory genes (*Il6, Nos2, Cox2,* and *Cxcl12*) were not expressed in Mφs at this stage, and the few that were expressed (*Ccl2* and *Cd86*) did not differ between genotypes (*Figure 1—figure supplement 4A*). To further investigate the effect of *Mmp14* inactivation on Mφ polarization, we activated BMDMs in vitro with lipopolysaccharide (LPS) or interleukin 4 (IL4) and analyzed the expression of M1- and M2-related genes, respectively (*Figure 1—figure supplement 4B*). No differences in Mφ polarization were found between WT and KO BMDMs, suggesting that *Mmp14* inactivation does not affect the Mφ M1/M2 activation spectrum.

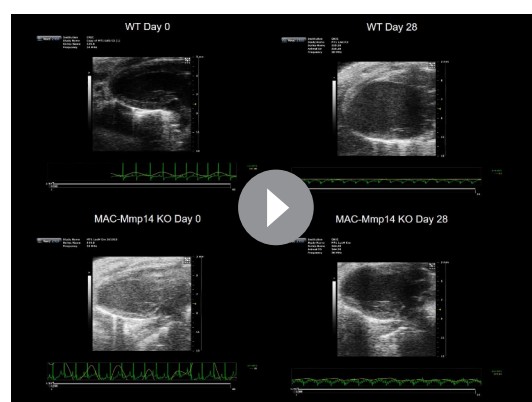

**Video 1.** Parasternal 2D long axis echocardiography view of WT (top) or MAC-Mmp14 KO (bottom) hearts at baseline (Day 0, left) and at 28 days post-MI (right) induced by LAD-ligation.
https://elifesciences.org/articles/57920#video1

## MAC-Mmp14 KO mice have a preserved microvasculature network and better myocardial oxygenation after ischemic injury

Next, we evaluated confocal images of WT and MAC-Mmp14 KO 7-day-post-MI hearts stained for CD31 (endothelial marker) and SMA (smooth muscle actin) with a fully automated 3D pipeline (Materials and methods), which allows reconstruction and quantification of the microvasculature (*Gkontra et al., 2018*; *Table 3*). MAC-Mmp14 KO 7-day-post-MI hearts had a higher vascular volume density and more capillaries and ECs within the infarction than WT hearts (*Figure 2A–B*). Oxygen diffusion from blood to tissue critically depends on the density and arrangement of the microvascular bed. Accordingly, in MAC-Mmp14 KO hearts, we observed increased capillary density and reduced intercapillary and diffusion distances, parameters proposed as indices of oxygen diffusion (*Gkontra et al., 2018*), suggesting better oxygen diffusion in the infarcted myocardium (*Figure 2B*). Next, we assessed hypoxia in the post LAD-ligation myocardium by carbonic anhydrase IX (CA-IX) staining (*Figure 2C*), which revealed a smaller CA-IX+ area in the infarction of 7-day-post-MI MAC-Mmp14 KO hearts than in WT counterparts (*Figure 2C–D*), confirming better-preserved tissue oxygenation after MI. No hypoxia signal was found in the non-infarcted tissue (*Figure 2—figure supplement 1A–B*). Although there were no between-genotype differences in arteriole (SMA$^+$ vessels) density, there was a trend toward thinner arteriole vessel walls at 7 days post-MI in MAC-Mmp14 KO infarcted hearts (*Figure 2—figure supplement 1C–D*), suggesting amelioration of MI-induced arterial hyperplasia (*Krishnamurthy et al., 2009*) in the absence of Mφ MT1-MMP.

## The inactivation of Mφ MT1-MMP impairs active TGFβ1 release from the LAP-TGFβ1 complex

To elucidate the mechanism by which Mφ MT1-MMP regulates microvascular density and fibrosis, we sought soluble factors whose release might be affected by MT1-MMP. Two independent proteomics studies previously performed in our laboratory detected a 1.25- to 2-fold increase in TGFβ1 content in the membrane-enriched subcellular fraction of *Mmp14* KO BMDMs. The small latent complex (SLC) consisting of latency associated peptide (LAP) and mature TGFβ1 can be retained at the cell surface through LAP binding to membrane receptors (e.g. integrin αvβ8) (*Figure 3A*). It has been shown that MT1-MMP promotes TGFβ1 activation via integrin αvβ8 at the cell surface (*Mu et al., 2002*), and that TGFβ1 modulates myocardial healing through effects on the fibrotic and angiogenic responses (*Frangogiannis, 2020*).

To investigate the possible role of Mφ-derived MT1-MMP in TGFβ1 release after MI, we used flow cytometry to measure latent TGFβ1 (LAP-TGFβ1 complex) on the surface of LPS-stimulated WT and MAC-Mmp14 KO BMDMs (*Figure 3A–C*). KO Mφs had a higher content of LAP and TGFβ1, revealing significantly higher surface retention of LAP-TGFβ1 than observed in WT Mφs (*Figure 3B–C*). To validate this observation, we detected TGFβ1 by immunoblot in the membrane and cytosolic subcellular fractions and total cell lysates of LPS-stimulated WT and MAC-Mmp14 KO BMDMs (*Figure 3D*). Latent TGFβ1 (~50 kDa) consisting of LAP (35 kDa) covalently bound to the mature TGFβ1 (12.5 kDa) preferentially located in the membrane and was more abundant in *Mmp14*-deficient Mφs (*Figure 3C–D*). In line with the reported role of MT1-MMP in the posttranslational processing of pro-TGFβ1, we found no difference in *Tgfb1* transcript levels between WT and MAC-Mmp14 KO BMDMs (*Figure 3E*).

Next, we queried whether impaired processing of LAP-TGFβ1 complex by *Mmp14*-deficient Mφs affects the secretion of active TGFβ1. Since active TGFβ1 has an estimated half-life of 2–3 min, and only 2–5% of total TGFβ1 is thought to be activated at any given time (*Lawrence, 2001*), we measured bioactive TGFβ1 using a standardized luciferase assay (Materials and methods). Responsiveness of the luciferase construct to TGFβ1 was confirmed by treating transfected HEK293 cells with TGFβ1 (2.94 ± 0.06 fold-increase in arbitrary luciferase units (ALU) with respect to the control). Luciferase activity was then assessed in transfected HEK293 cells cultured in the presence or absence of LPS-stimulated BMDMs for 24 hr (*Figure 3F*). HEK293 co-culture with WT BMDMs produced abundant levels of active TGFβ1, whereas active TGFβ1 was undetectable after co-culture with MAC-Mmp14 KO BMDMs (*Figure 3F*). TGFβ1 levels in MAC-Mmp14 KO BMDMs were restored by transduction with full-length (FL) MT1-MMP but not with a catalytic mutant (E240A), demonstrating that MT1-MMP-dependent TGFβ1 activation requires MT1-MMP catalytic activity (*Figure 3G*).

**Table 3.** Quantitative analysis of microvasculature parameters in infarcted cardiac tissue from WT and MAC-Mmp14 KO mice on day 7 after MI.

Capillaries correspond to CD31$^+$SMA$^-$ vessels of diameter <3 μm. Data are means ± SEM of 5–6 mice per genotype. Unpaired *t*-test. Significant differences are indicated as *p<0.05.

| Minkowski-based metrics | WT | MAC-Mmp14 KO |
|---|---|---|
| Vascular volume density (%) | 12.65 ± 0.36 | 13.26 ± 0.34 * |
| Vascular surface area density (× 10$^{-3}$) (μm$^2$/μm$^3$) | 21.24 ± 2.5 | 23.13 ± 1.92 |
| *Graph-based metrics* | | |
| Vascular segment length (μm) | 6.97 ± 0.74 | 6.2 ± 0.18 |
| Vascular segment surface (μm$^2$) | 36.41 ± 5.52 | 29.5 ± 1.6 |
| Vascular segment volume (μm$^3$) | 17 ± 3.8 | 12.54 ± 1.53 |
| Tortuosity (μm/μm) | 1.61 ± 0.02 | 1.61 ± 0.03 |
| Vascular segments (× 10$^5$)[a] | 4.12 ± 0.68 | 5.48 ± 0.53 |
| Vascular segments[b] | 144.17 ± 14.15 | 158.27 ± 2.31 |
| Vessels of diameter <= 3 (μm) (%) | 95.48 ± 2.55 | 96.97 ± 1.85 * |
| Vessels of diameter between 3 and 6 (μm) (%) | 4.51 ± 2.52 | 3.03 ± 1.85 |
| Vessels of diameter > 6 (μm) (%) | 0.012 ± 0.001 | 0.028 ± 0.001 |
| Vessels of diameter <= 3 (μm) (× 10$^5$)[a] | 3.97 ± 0.71 | 5.33 ± 0.59 * |
| Vessels of diameter between 3 and 6 (μm) (× 10$^5$)[a] | 0.16 ± 0.06 | 0.15 ± 0.09 |
| Vessels of diameter > 6 (μm)[a] | 32.81 ± 0.0001 | 73.36 ± 0.0001 |
| Branching nodes (× 10$^4$)[a] | 22.39 ± 4.28 | 30.43 ± 2.51 |
| Blind-ends/sprouts (× 10$^4$)[a] | 6.06 ± 0.92 | 7.93 ± 1.2 |
| Branching nodes[b] | 77.38 ± 9.59 | 87.31 ± 2.14 |
| Blind-ends/sprouts[b] | 24.09 ± 4.07 | 25.9 ± 1.66 |
| *SMA-related metrics* | | |
| Vessels covered with SMA (%) | 47.2 ± 17.67 | 53.54 ± 12 |
| SMA$^+$ layer thikness (μm) | 2.98 ± 0.76 | 2.72 ± 0.43 |
| Damage index | 0.21 ± 0.11 | 0.19 ± 0.09 |
| Myofibroblasts (× 10$^4$)[a] | 2.51 ± 1.18 | 2.05 ± 0.64 |
| Myofibroblasts[b] | 8.9 ± 3.8 | 5.97 ± 1.47 |
| Myofibroblasts (× 10$^5$)[d] | 19.1 ± 10 | 15.3 ± 4.7 |
| SMA+ perivascular cells (× 10$^4$)[a] | 4.74 ± 2.36 | 5.25 ± 1.59 |
| SMA+ perivascular cells[b] | 16.16 ± 7.31 | 15.48 ± 3.65 |
| SMA+ perivascular cells (× 10$^5$)[d] | 36.3 ± 19.3 | 38.9 ± 10.8 |
| *Efficiency in oxygen diffusion* | | |
| Maximal extravascular distance (μm) | 51.47 ± 10.83 | 45.47 ± 7.11 |
| Median extravascular distance (μm) | 14.28 ± 2.64 | 13 ± 1.38 |
| Capillary density[c] | 1201 ± 298.12 | 1720 ± 368.61 * |
| Intercapillary distance | 8.09 ± 0.51 | 7.31 ± 0.27 * |
| Diffusion distance | 10.84 ± 0.51 | 9.76 ± 0.49 * |
| *Additional cell-related metrics* | | |
| Endothelial cells (× 10$^4$)[a] | 6.2 ± 1.05 | 7.91 ± 1.21 |
| Endothelial cells[b] | 22.45 ± 3.91 | 23.16 ± 1.34 |
| Endothelial cells (× 10$^5$)[d] | 46.6 ± 8.1 | 58.8 ± 7.7 * |

[a] per mm$^3$ of tissue, [b] per mm vessel length, [c] per mm$^2$ of tissue, [d] per mm$^3$ vessel volume.

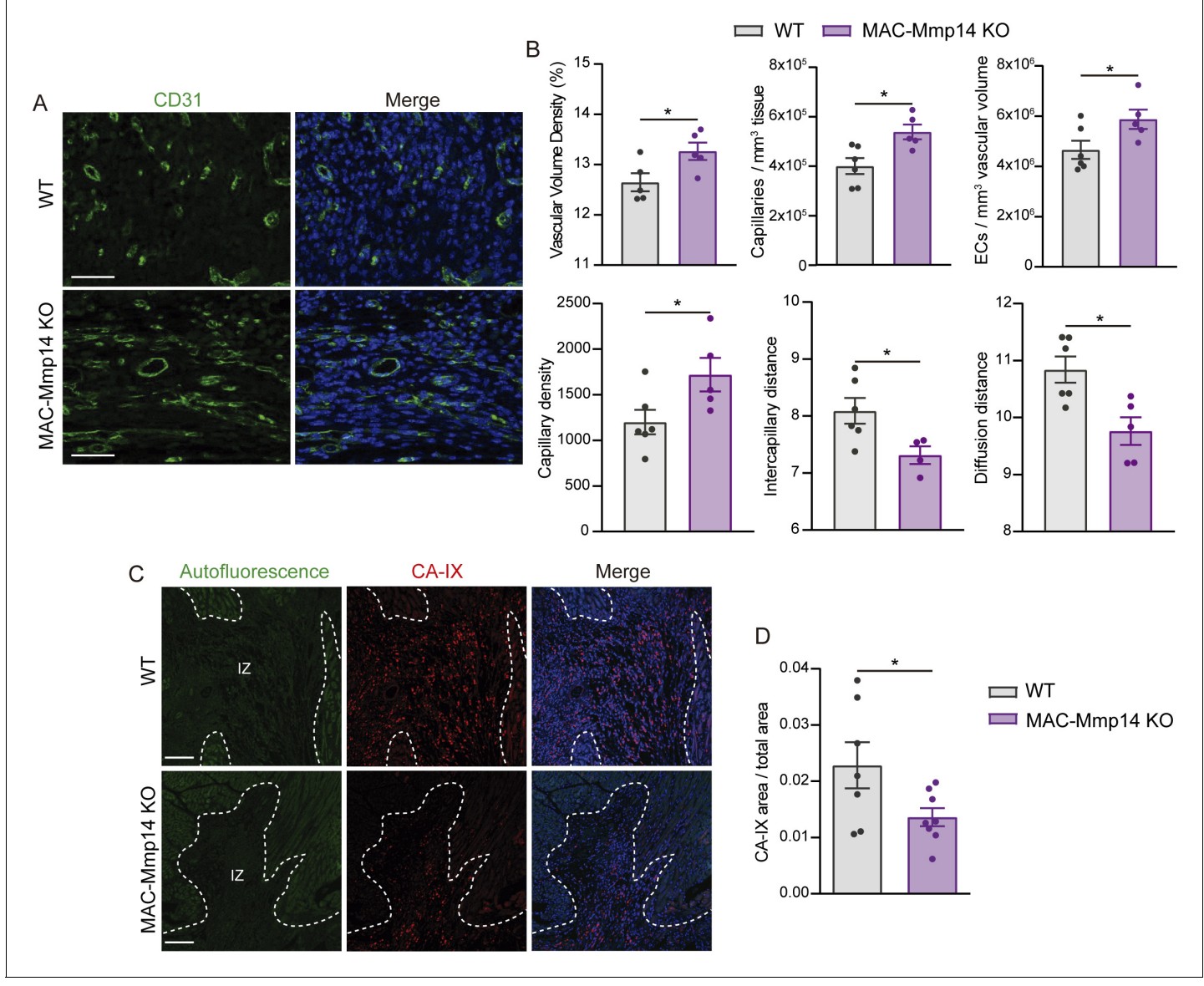

**Figure 2.** MAC-Mmp14 KO mice have a preserved microvasculature network and better myocardial oxygenation after ischemic injury. (A) Representative confocal microscopy images showing immunostaining for CD31 (green) and nuclei (blue) within the infarction in WT and MAC-Mmp14 KO hearts at 7 days post-MI. Scale bar, 50 μm. (B) Vasculature-related parameters within the infarction at 7 days post-MI. Data are means ± SEM of 5–6 mice per genotype. Unpaired $t$-test. (C) Representative confocal immunofluorescence microscopy images of CA-IX (red) in the infarcted region of WT and MAC-Mmp14 KO hearts at 7 days post-MI. Nuclei are stained with DAPI (blue). Scale bar, 100 μm. (D) CA-IX$^+$ area:total area ratio in the infarcted zone. Data are means ± SEM of 7–8 mice per genotype. Unpaired $t$-test.

The online version of this article includes the following source data and figure supplement(s) for figure 2:

**Source data 1.** MAC-Mmp14 KO mice have a preserved microvasculature network and better myocardial oxygenation after ischemic injury.
**Figure supplement 1.** Remodeling of cardiac vasculature in MAC-Mmp14 KO mice after MI.
**Figure supplement 1—source data 1.** Remodeling of cardiac vasculature in MAC-Mmp14 KO mice after MI.

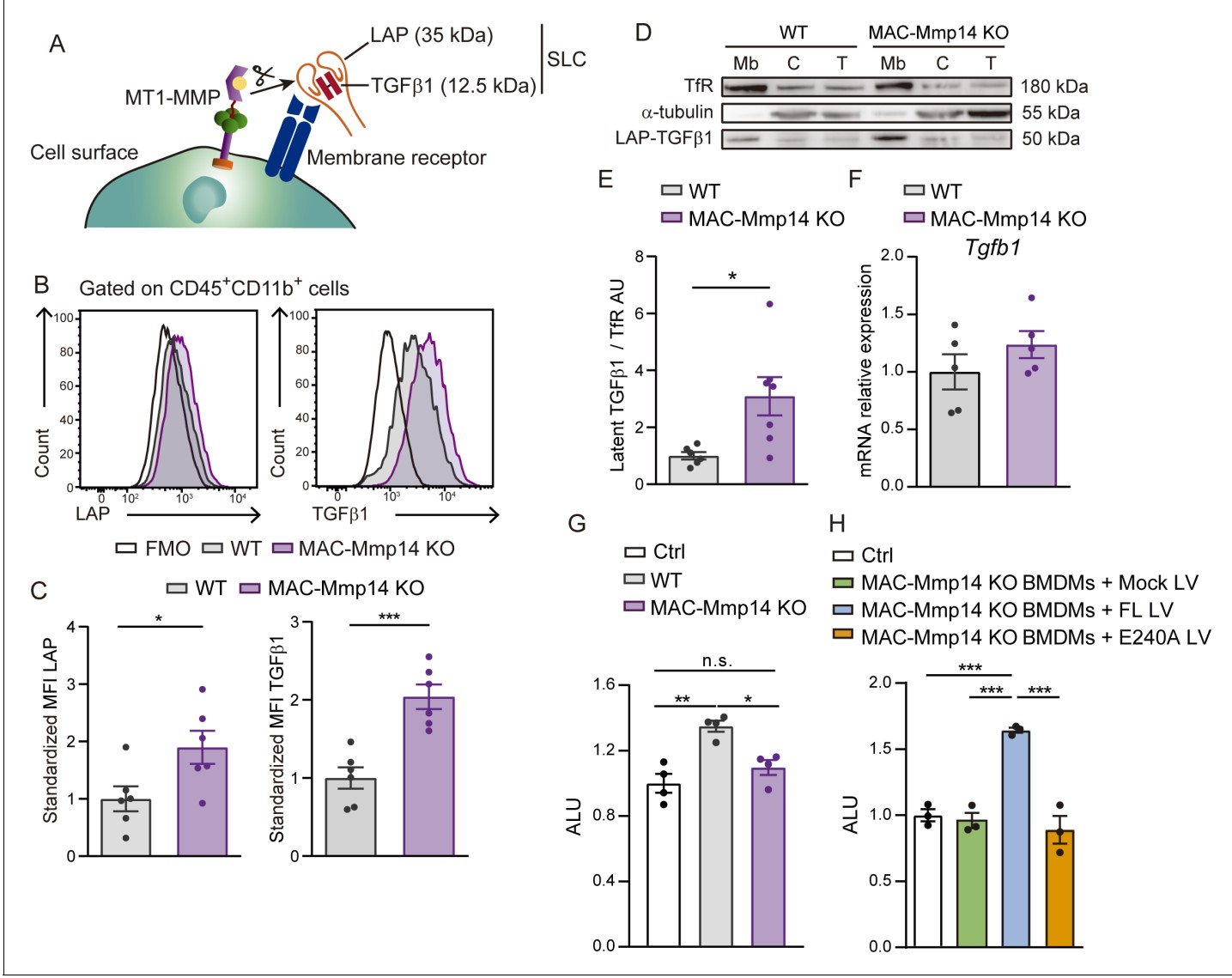

**Figure 3.** Mφ-inactivation of MT1-MMP impairs active TGFβ1 release from LAP-TGFβ1 complex. (**A**) Scheme of LAP-TGFβ1 complex retention in the cell surface through LAP-binding to membrane receptors. (**B**) Representative flow cytometry histogram plots of LAP and TGFβ1 staining in LPS-activated WT and MAC-Mmp14 KO BMDMs. (**C**) Standardized mean fluorescence intensity (MFI) of LAP and TGFβ1 in experiments as in B. Data are means ± SEM of seven mice per group. Unpaired *t*-test. (**D**) Western blot of transferrin receptor (TfR), α-tubulin, and LAP-TGFβ1 complex in membrane fraction (Mb), cytosolic fraction (C), and total lysate (T) from LPS-activated WT and MAC-Mmp14 KO BMDMs. (**E**) Quantification of LAP-TGFβ1 complex in the membrane fraction. Data are means ± SEM of 6–7 mice per genotype. Unpaired *t*-test. (**F**) *Tgfb1* mRNA expression in LPS-activated WT and MAC-Mmp14 KO BMDMs. Data are means ± SEM of five mice per genotype. Unpaired *t*-test. (**G**) Arbitrary luciferase units (ALU) in HEK293 cells co-cultured with or without LPS-activated WT or MAC-Mmp14 KO BMDMs. Control corresponds to transfected HEK293 cells cultured alone. Data are means ± SEM of a representative experiment of three performed with four technical replicates per condition. One-way ANOVA followed by Tukey's multiple comparisons test. (**G**) ALU in HEK293 cells co-cultured with or without conditioned media from LPS-activated MAC-Mmp14 KO BMDMs transduced with mock lentivirus (GFP), or lentivirus containing full-length MT1-MMP (FL) or catalytic MT1-MMP mutant (E240A). Control corresponds to transfected HEK293 cells cultured alone. Data are means ± SEM of a representative experiment of three performed with three technical replicates per condition. One-way ANOVA followed by Tukey's multiple comparisons test.

The online version of this article includes the following source data for figure 3:

**Source data 1.** Mφ-inactivation of MT1-MMP impairs active TGFβ1 release from the LAP-TGFβ1 complex.

## The inactivation of Mφ MT1-MMP reduces TGFβ1-pSMAD2 signaling in cardiac ECs, MyoFBs, and VSMCs after MI

We explored if the defective release of active TGFβ1 by *Mmp14*-deficient Mφs could affect TGFβ1-mediated SMAD2/3 in the myocardium. We quantified phosphorylated SMAD2 (pSMAD2) in Mφs, ECs, and myofibroblasts (MyoFBs) by flow cytometry in 7-day-post-MI hearts (*Figure 4A–B*). Mφs were defined as CD45$^+$CD11b$^+$F4/80$^+$ cells, and ECs as CD45$^-$CD31$^+$ cells (*Figure 4A*). Cardiac FBs convert to MyoFBs with injury to mediate healing after MI and platelet-derived growth factor receptor β (PDGFRβ) induction is an early feature of MyoFB activation (*Henderson et al., 2013*). In line with our previous papers (*Gkontra et al., 2018*; *Żak et al., 2019*), histological examination revealed that PDGFRβ indeed labels two distinct cell subsets in the infarcted myocardium: (i) PDGFRβ$^+$ cells around vessels (pericytes and vascular smooth muscle cells (VSMCs)), and (ii) non-perivascular PDGFRβ$^+$ cells (MyoFBs) (*Figure 4—figure supplement 1*). We, therefore, identified MyoFBs and VSMCs as CD45$^-$CD31$^-$PDGFRβ$^+$ cells (*Figure 4A*). FACS of CD45$^-$CD31$^-$PDGFRβ$^+$ cells followed by qPCR analysis confirmed the expression of MyoFB markers such as *Tagln*, *Col1a1*, *Col1a2*, and *Col1a3* (*Figure 5—figure supplement 1A–B*). There was no between-genotype difference in pSMAD2 abundance in Mφs; however, pSMAD2 was significantly less abundant in ECs and MyoFBs/VSMCs from MAC-Mmp14 KO hearts (*Figure 4B–C*), suggesting that lack of Mφ MT1-MMP impairs paracrine TGFβ1-pSMAD2 signaling in ECs and MyoFBs/VSMCs. To distinguish between pSMAD2 in MyoFBs (non-vascular-related SMA$^+$ cells) and VSMCs (vascular-related SMA$^+$ cells), we performed confocal imaging analysis at 7 days post-MI which revealed that pSMAD2-positive MyoFBs and VSMCs were both less abundant in MAC-Mmp14 KO hearts than in WT hearts (*Figure 4D–E*).

We also investigated the possible effect on cardiac FBs and CMs of reduced TGFβ1 release by *Mmp14*-deficient Mφs. qPCR analysis of cardiac FBs and CMs after culture with supernatants from LPS-activated WT or MAC-Mmp14 KO BMDMs revealed no changes except for the direct TGFβ1 gene target *Serpine1*, whose expression was significantly higher in FBs treated with conditioned medium from WT BMDMs (*Figure 4—figure supplement 2A*). This result is consistent with the higher production of active TGFβ1 by WT BMDMs than by MAC-Mmp14 KO BMDMs (*Figure 3G*) and the higher abundance of MyoFBs in KO hearts, suggesting a role for MT1-MMP in FB activation to MyoFBs.

## Mφ-derived MT1-MMP induces EndMT after MI

We found that Mφ MT1-MMP inactivation altered the cellular composition of the infarcted myocardium, with MAC-Mmp14 KO mice showing fewer Mφs, indicating a less inflammatory state; more ECs, in line with the microvasculature image analysis data; and fewer MyoFBs, in agreement with the reduced fibrotic response (*Figure 5A–B*). Additionally, we detected an intermediary population of cells with mild levels of both CD31 (endothelial marker) and PDGFRβ (mesenchymal marker) (*Figure 5A*). This population is suggestive of transitioning cells undergoing endothelial to mesenchymal transition (EndMT) (*Aisagbonhi et al., 2011*). To confirm the occurrence of EndMT in the context of MI, we performed lineage tracing experiments in *Cdh5*-Cre$^{ERT2}$ R26Tomato control and infarcted mice and asked whether we could find endothelial-cell derived Tomato$^+$ cells within the MyoFB compartment using flow cytometry (*Figure 5—figure supplement 2A–E*). We detected an increase in Tomato$^+$PDGFRβ$^+$ cells in 7-day-post-MI hearts (*Figure 5—figure supplement 2D,E*), confirming that EndMT is triggered upon MI as previously reported (*Aisagbonhi et al., 2011*). We confirmed the transitioning phenotype of CD31$^+$PDGFRβ$^+$ cells by qPCR and observed the loss of endothelial markers (i.e. *Pecam*, *Cdh5*, *Kdr*, *Tie2*, *Col4a2*), the acquisition of mesenchymal genes (i.e. *Cdh2*, *Tagln*, *Col1a1*, *Col1a2*, *Col3a1*), and the upregulation of EndMT-mediating transcriptional factors (i.e. *Zeb2* and *Snai1*) (*Figure 5—figure supplement 1A–B*). Interestingly, these cells were less abundant in MAC-Mmp14 KO hearts (*Figure 5A–B*), suggesting that EndMT is impaired in the absence of Mφ MT1-MMP. We confirmed the results obtained with PDGFRβ staining by the complementary staining for MEFSK4 (*Figure 5—figure supplement 3A,B*), another marker for cardiac FB lineage (*Pinto et al., 2016*).

TGFβ is considered the master mediator of EndMT, in a SMAD-dependent manner (*Frangogiannis, 2020*). We found by immunofluorescence triple-positive cells for the endothelial nuclear marker ERG, the mesenchymal marker SMA and pSMAD2, reflecting the transition of endothelial cells towards a mesenchymal phenotype via SMAD2 activation in the infarcted cardiac tissue (*Figure 5—*

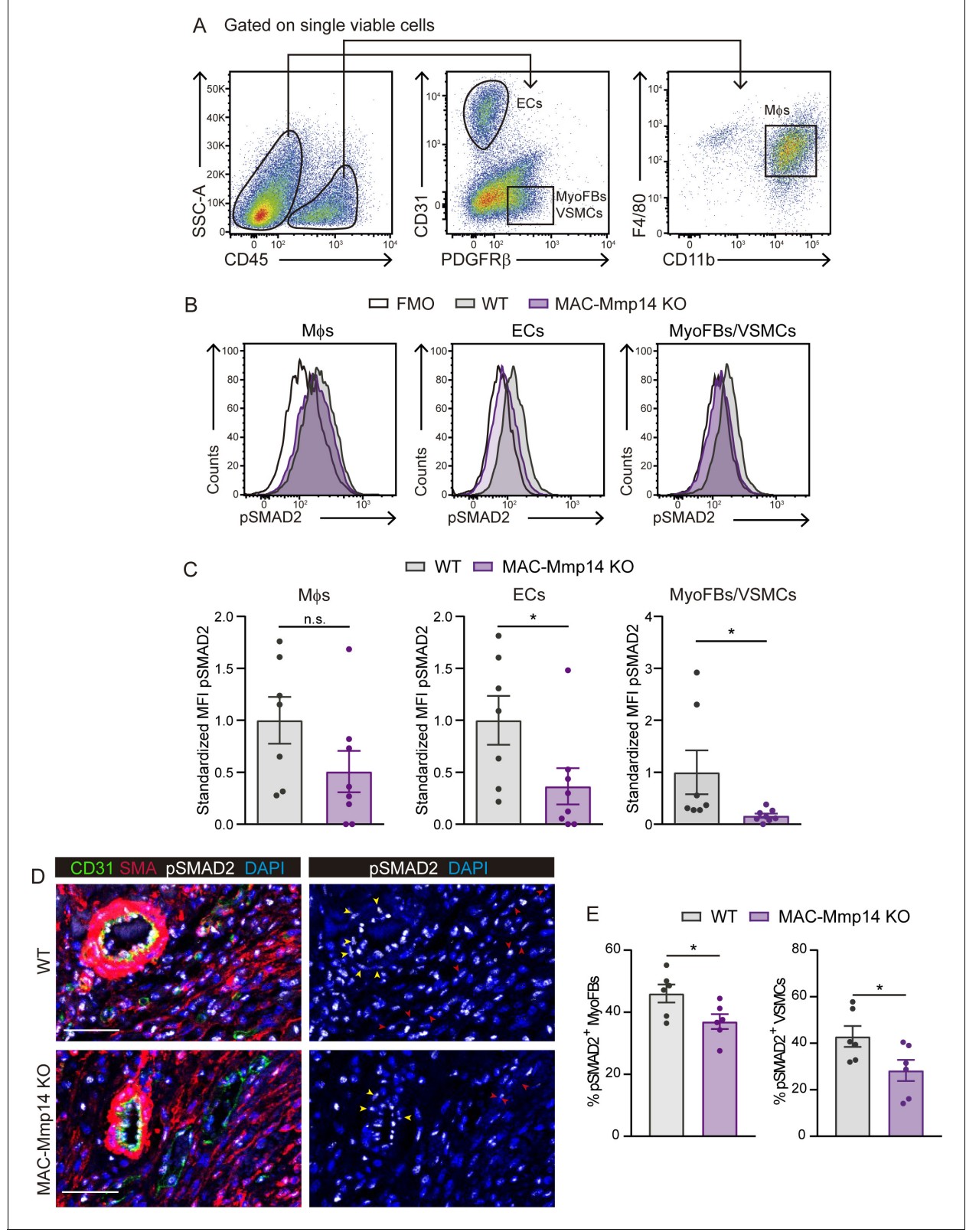

**Figure 4.** The inactivation of Mφ MT1-MMP reduces TGFβ1-pSMAD2 signaling in cardiac ECs, MyoFBs, and VSMCs after MI. (**A**) Gating strategy used to assess pSMAD2 signaling in ECs, MyoFBs/VSMCs, and Mφs. (**B**) Representative flow cytometry histogram plots of pSMAD2 staining in the indicated cells from 7-day-post-MI hearts. (**C**) Standardized MFI of pSMAD2 in experiments as in B. Data are means ± SEM of 7–8 mice per genotype. Unpaired *t*-test. (**D**) Representative immunofluorescence staining of CD31 (green), SMA (red), and pSMAD2 (white) in infarcted cardiac tissue from WT mice (top)
*Figure 4 continued on next page*

*Figure 4 continued*

and MAC-Mmp14 KO mice (bottom) at 7 days post-MI. Nuclei are stained with DAPI (blue). Red and yellow arrowheads point to pSMAD2⁺ MyoFBs and pSMAD2⁺ VSMCs, respectively. Scale bar, 50 μm. (**E**) Percentages of pSMAD2⁺ MyoFBs and pSMAD2⁺ VSMCs within the total MyoFB or VSMC populations, respectively in the infarcted zone. Data are means ± SEM of six mice per genotype. Unpaired *t*-test.

The online version of this article includes the following source data and figure supplement(s) for figure 4:

**Source data 1.** The inactivation of Mφ MT1-MMP reduces TGFβ1-pSMAD2 signaling in cardiac ECs, MyoFBs, and VSMCs after MI.
**Figure supplement 1.** PDGFRβ is expressed by cardiac MyoFBs and VSMCs.
**Figure supplement 2.** Effect of WT or MAC-Mmp14 KO Mφs on FBs and CMs.
**Figure supplement 2—source data 1.** Effect of WT or MAC-Mmp14 KO Mφs on FBs and CMs.

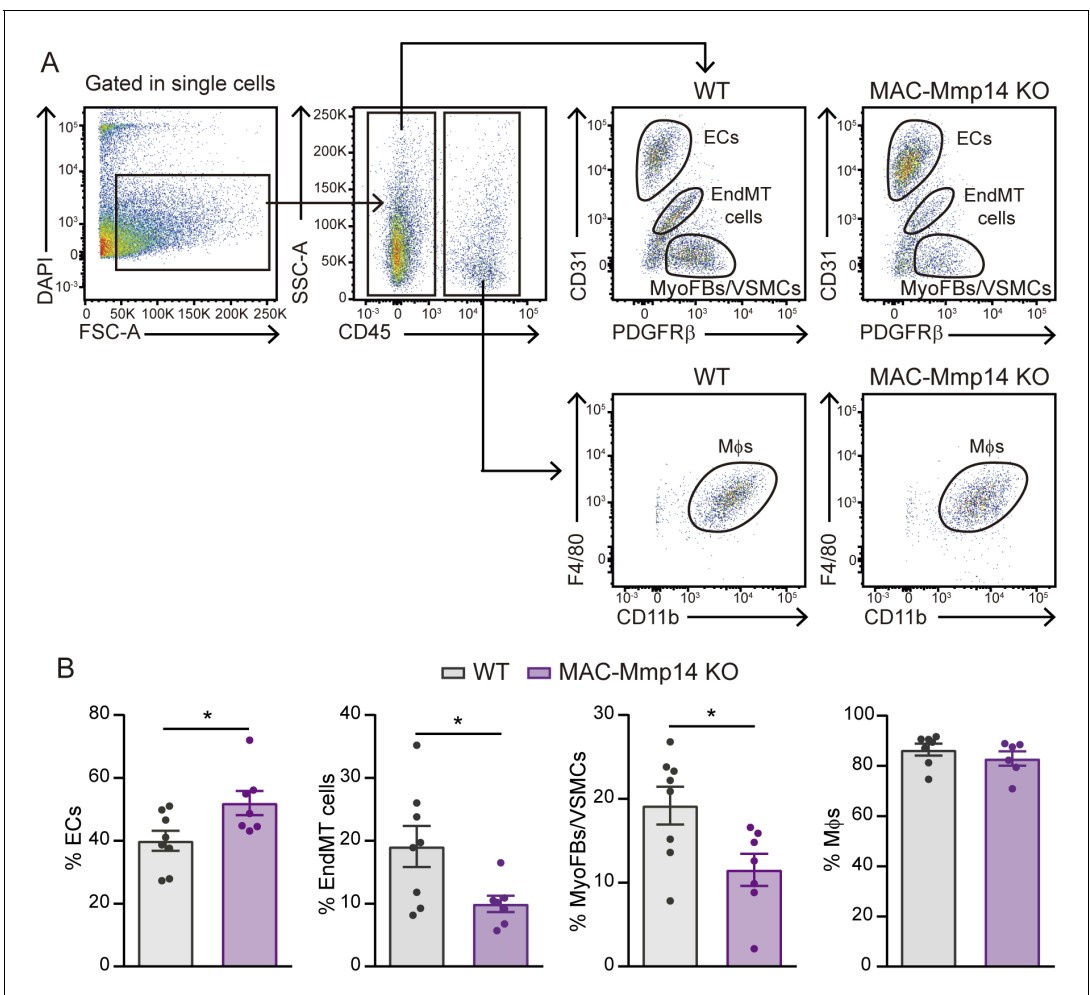

**Figure 5.** The inactivation of Mφ MT1-MMP alters myocardial cellular composition after MI. (**A**) Flow cytometry gating strategy used to identify and quantify cardiac ECs, MyoFBs, cells undergoing EndMT, and Mφs in WT mice (left) and MAC-Mmp14 KO mice (right) on day 7 after MI. (**B**) Quantification of Mφs, ECs, MyoFBs, and cells undergoing EndMT in cardiac tissue 7 days after MI. Data are means ± SEM of at least 11 mice per genotype. Unpaired *t*-test.

The online version of this article includes the following source data and figure supplement(s) for figure 5:

**Source data 1.** The inactivation of Mφ MT1-MMP alters myocardial cellular composition after MI.
**Figure supplement 1.** Endothelial-to-mesenchymal gene signature of CD31⁺PDGFRβ⁺ cells.
**Figure supplement 1—source data 1.** Endothelial-to-mesenchymal gene signature of CD31⁺PDGFRβ⁺ cells.
**Figure supplement 2.** Lineage tracing of endothelial derived-mesenchymal cells.
**Figure supplement 3.** The inactivation of Mφ MT1-MMP attenuates post-MI EndMT.
**Figure supplement 3—source data 1.** The inactivation of Mφ MT1-MMP attenuates post-MI EndMT.

*figure supplement 3C–D*). We, therefore, studied the effect on post-MI EndMT of the impaired TGFβ1 production by *Mmp14*-deficient Mφs, and the associated reduction in paracrine pSMAD2 signaling in ECs. For that, we co-cultured purified mouse aortic endothelial cells (MAECs) with LPS-activated WT or MAC-Mmp14 KO BMDMs (*Figure 6—figure supplement 1*). The immunofluorescence analysis with the endothelial marker CD31 and the mesenchymal marker SMA revealed morphological changes in MAECs after co-culture with WT BMDMs, with cells progressively losing their cobblestone appearance and adopting a dispersed, spindle-shaped morphology (*Figure 6—figure supplement 1A*). Moreover, MAECs co-cultured with WT BMDMs significantly decreased CD31 expression and acquired SMA expression, indicating the acquisition of molecular traits of a mesenchymal phenotype (*Figure 6—figure supplement 1A–B*). This phenotype was similar to the effects of recombinant TGFβ1 stimulation (*Figure 6—figure supplement 1A–B*). By contrast, co-culture of MAECs with MAC-Mmp14 KO BMDMs led neither to the loss of endothelial features nor to the acquisition of mesenchymal markers (*Figure 6—figure supplement 1A–B*). Treatment of MAEC-BMDM co-cultures with a neutralizing anti-TGFβ1 antibody blocked EndMT, demonstrating that Mφs induced EndMT via MT1-MMP/TGFβ1 (*Figure 6—figure supplement 1C–D*). Altered expression of endothelial and mesenchymal markers in the co-cultures was confirmed by qPCR. In accordance with the immunofluorescence data, the co-culture of MAECs with WT BMDMs caused the downregulation of EC-related genes (*Pecam*, *Kdr*, and *Col4a2*) and the upregulation of mesenchymal genes (*Tagln* and *Acta2*) and the direct TGFβ1 target *Serpine1* (*Pai1*). A similar phenotype was induced by TGFβ1-treatment. By contrast, these changes toward a mesenchymal phenotype in MAECs were not triggered by co-culture with MAC-Mmp14 KO BMDMs (*Figure 6—figure supplement 2*).

To confirm Mφ MT1-MMP-dependent induction of EndMT in vivo following MI, we first investigated TGFβ1 processing in cardiac Mφs. LAP and TGFβ1 were both significantly increased on the surface of MAC-Mmp14 KO Mφs after MI, indicating the retention of latent TGFβ1 (*Figure 6A*). We then sorted cardiac Mφs from 7-day-post-MI WT and MAC-Mmp14 KO hearts (*Figure 1—figure supplement 1A–B*) and co-cultured them with luciferase-transfected HEK293 cells. As with BMDMs, post-MI WT cardiac Mφs produced detectable levels of active TGFβ1, whereas the inactivation of *Mmp14* in Mφs abrogated TGFβ1 activation (*Figure 6B*). In co-culture experiments, WT cardiac Mφs induced EndMT in MAECs, downregulating CD31 expression, and increasing SMA expression. By contrast, no transition to a mesenchymal phenotype was evident in MAECs co-cultured with *Mmp14*-deficient cardiac Mφs (*Figure 6C–D*).

## Discussion

MI results in loss of CMs, adverse structural remodeling, and LV dysfunction and dilation, eventually causing heart failure. Since appropriate cardiac repair requires a balanced inflammatory response to avoid adverse cardiac remodeling after MI, Mφs have emerged as likely candidates for investigation and therapeutic intervention. Our results show that post-MI Mφs have heightened expression of *Mmp14* as well as its substrates *Mmp2* and *Col1a1*, in line with a role in tissue remodeling and collagen deposition (*O'Rourke et al., 2019*). Although ECM remodeling and granulation tissue formation are prerequisites for tissue repair, excessive MMP activity after MI and subsequent ECM turnover can result in adverse remodeling and worsened cardiac dysfunction (*Spinale et al., 2010*). Therefore, we hypothesized that Mφ-specific inactivation of MT1-MMP might limit adverse remodeling and LV dilation and dysfunction after MI.

Although MAC-Mmp14 KO mice are phenotypically normal under homeostatic conditions, these animals were protected when challenged by acute MI. MAC-Mmp14 KO mice had smaller infarcts and better LV contractility after MI than WT mice, as well as significantly improved preservation of systolic function and LV structure. A detrimental role of MT1-MMP in post-MI cardiac remodeling has been identified using mouse infarct models, leading to worsening cardiac function and reduced survival (*Koenig et al., 2012*; *Spinale et al., 2010*; *Zavadzkas et al., 2011*). These studies attributed this detrimental role to MT1-MMP collagenase activity in FBs, disregarding its actions in Mφs (*Koenig et al., 2012*). Our data obtained with the MAC-Mmp14 KO model are the first demonstrating the beneficial effect of Mφ-specific MT1-MMP inactivation in preventing adverse LV remodeling after MI. This effect is likely caused by the lower myocardial fibrosis, allowing the heart to work at less of a mechanical disadvantage, and better oxygenation of the infarcted myocardium due to the preservation of the microvasculature network.

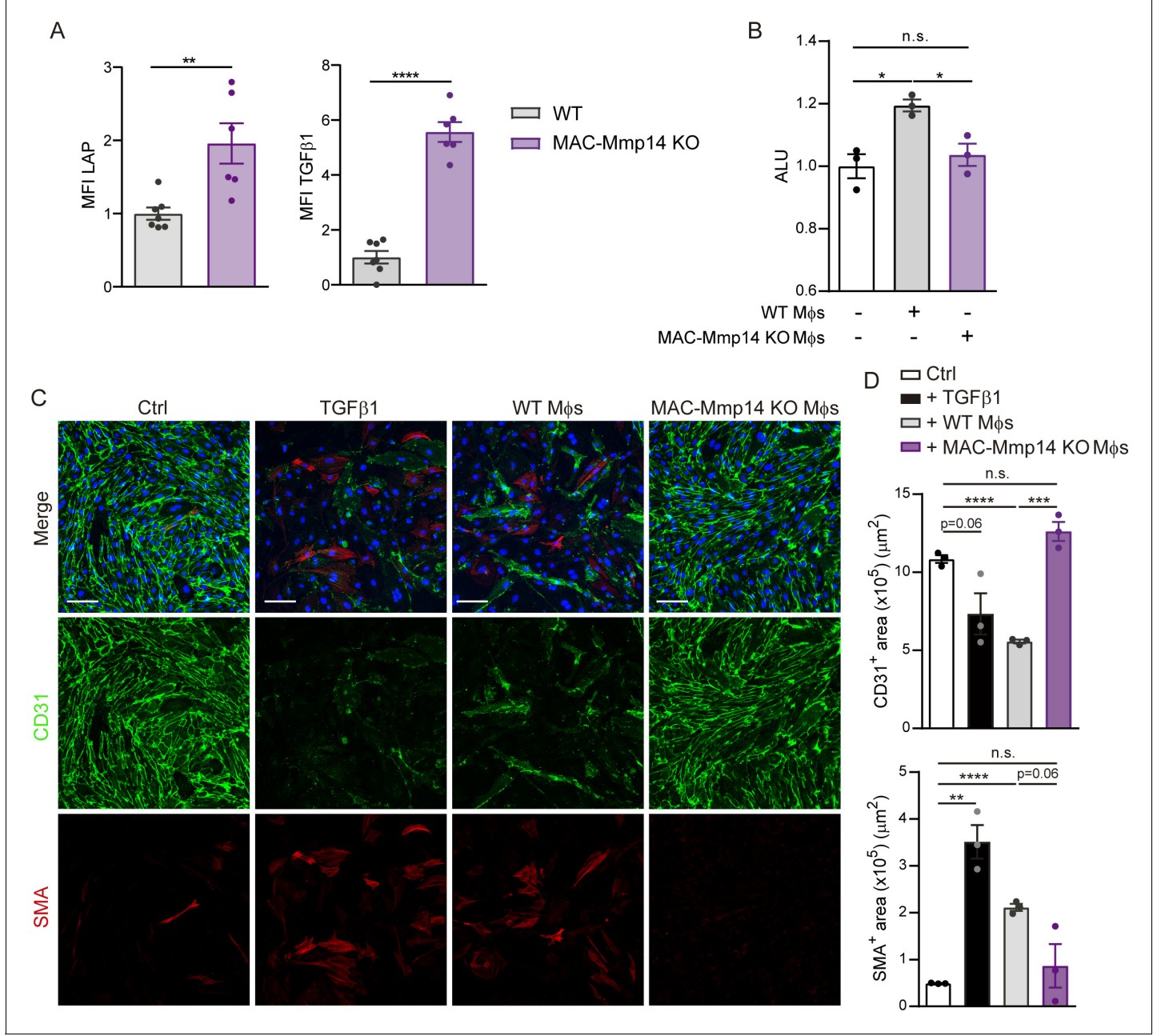

**Figure 6.** Cardiac Mφs induce post-MI EndMT through MT1-MMP-mediated TGFβ1 activation. (**A**) Standardized MFI of LAP and TGFβ1 staining in WT and MAC-Mmp14 KO cardiac Mφs on day 7 after MI. Data are means ± SEM of 6–7 mice per group. Unpaired *t*-test. (**B**) Luciferase activity (ALU) in transfected HEK293 cells co-cultured with Mφs from 7-day-post-MI WT or MAC-Mmp14 KO hearts. Data are means ± SEM of three independent experiments performed with four technical replicates per condition. One-way ANOVA followed by Tukey's multiple comparisons test. (**C**) Representative immunofluorescence staining of CD31 (green) and SMA (red) in in vitro co-cultures of MAECs and cardiac Mφs from WT or MAC-Mmp14 KO 7-day-post-MI hearts. Nuclei are stained with DAPI (blue). Scale bar, 100 µm (**D**) CD31+ area (µm2) and SMA+ area (µm2) in the different conditions. Data are means ± SEM of a representative experiment of three performed with three technical replicates per condition. Unpaired *t*-test. The online version of this article includes the following source data and figure supplement(s) for figure 6:

**Source data 1.** Cardiac Mφs induce post-MI EndMT through MT1-MMP-mediated TGFβ1 activation.

**Figure supplement 1.** Mφs induce EndMT in vitro via MT1-MMP/TGFβ1.

**Figure supplement 1—source data 1.** Mφs induce EndMT in vitro via MT1-MMP/TGFβ1.

**Figure supplement 2.** MT1-MMP is required for in vitro Mφ induction of EndMT.

**Figure supplement 2—source data 1.** MT1-MMP is required for in vitro Mφ induction of EndMT.

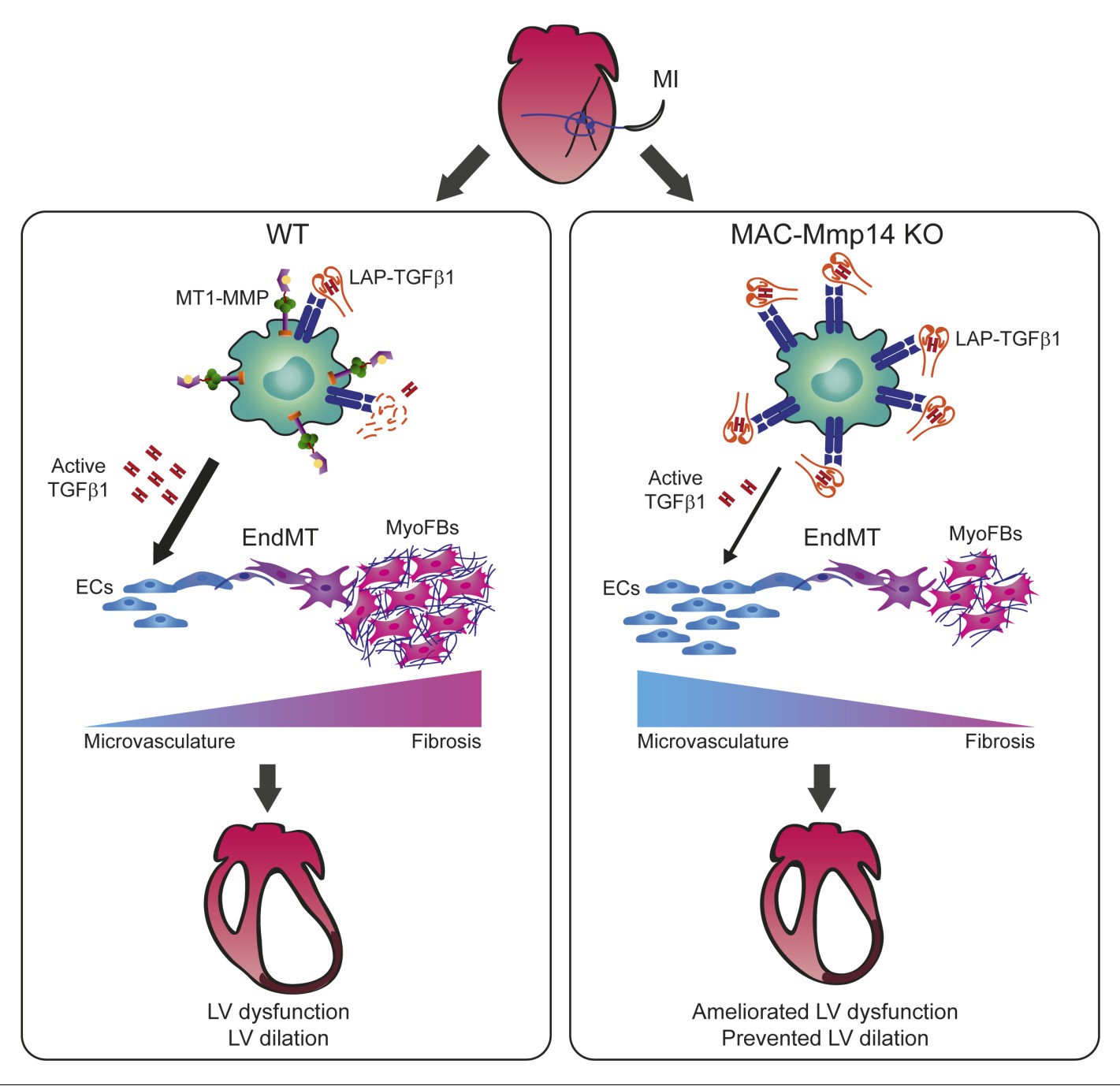

**Figure 7.** Mφ-inactivation of MT1-MMP preserves cardiac function after MI by impairing TGFβ1-mediated EndMT. MI triggers MT1-MMP production by Mφs, contributing to the release of active TGFβ1 from SLC (LAP-TGFβ1) to the myocardium. Active TGFβ1 signals acting on ECs promote EndMT, contributing to adverse tissue remodeling. When Mφ MT1-MMP is absent, latent TGFβ1 accumulates, and the availability of active TGFβ1 in the myocardium decreases. In this scenario, the impairment of Mφ-mediated EndMT results in enhanced angiogenesis and reduced fibrosis, limiting LV remodeling, and preserving cardiac function.

Loss of collagenase activity in MAC-Mmp14 KO mice might be expected to lead to a dense scar. However, in addition to its collagenolytic activity (*Ohuchi et al., 1997*), MT1-MMP proteolytically processes a diverse range of biologically active signaling molecules (*Koziol et al., 2012*); stimulation of collagen synthetic pathways by these molecules would explain the observed phenotype of post-MI MAC-Mmp14 KO mice. Pro-TGFβ has been recognized as a target for MT1-MMP–mediated

cleavage in several contexts (*Koziol et al., 2012*; *Mu et al., 2002*). Our results provide evidence for MT1-MMP-mediated activation of latent LAP-TGFβ1 complex in cardiac Mφs after MI. Dampened processing and release of active TGFβ1 in MAC-Mmp14 KO mice was associated with a decrease in SMAD2-mediated signaling in the infarcted myocardium, providing a mechanism for the reduced pro-fibrotic response and preservation of the microvasculature network. Previous studies identified MT1-MMP as an inducer of fibrosis through its cleavage of latent-transforming growth factor beta-binding protein 1 (LTBP1) and activation of TGFβ-mediated SMAD2/3 signaling in the infarcted myocardium (*Spinale et al., 2010*; *Zavadzkas et al., 2011*). TGFβ1 is the main driver of FB differentiation to MyoFBs (*van den Borne et al., 2010*); our results, showing reduced MyoFBs numbers in MAC-Mmp14 KO hearts and depressed *Serpine1* expression by FBs treated with MAC-Mmp14 KO Mφs, suggest that MT1-MMP plays a role in MyoFB activation via TGFβ1. Suppressed TGFβ1 production may also underlie the preservation of the microvasculature network in MAC-Mmp14 KO mice, through the loss of TGFβ1 angiostatic effects (*Arnold et al., 2014*; *Imaizumi et al., 2010*). However, the reduced release of TGFβ1 by MAC-Mmp14 KO Mφs did not appear to affect, autocrine or paracrine, the Mφ polarization phenotype or the gene signature of CM and FB (other than *Serpine1*). Further studies will be needed to fully define these processes.

TGFβ-mediated EndMT has been identified during cardiac fibrosis (*Zeisberg et al., 2007*) and contributes to collagen matrix deposition and disease. The concomitant loss of functional ECs may also lead to capillary rarefaction, thus causing tissue ischemia, a potent driver of fibrosis. EndMT is essential for cardiac valve formation and vascular development in embryogenic stages as well as in the pathogenesis of diverse cardiovascular disorders, such as congenital heart disease (*Hofmann et al., 2012*; *Xu et al., 2015*). The contribution of EndMT to cardiac fibrosis remains however a matter of debate, which depends on the nature of the cardiovascular injury and the extent of the fibrosis (*Aisagbonhi et al., 2011*; *Evrard et al., 2016*; *Kanisicak et al., 2016*; *Moore-Morris et al., 2014*; *Zeisberg et al., 2007*). For example, EndMT is minimal in pressure overload or I/R models, where the intensity of inflammatory and fibrotic responses is much lower than after acute ischemic injury (*Moore-Morris et al., 2014*; *Xia et al., 2009*). Besides, the tools employed for assessing EndMT present several limitations; for instance, immunofluorescence techniques are not sensible enough to reliably detect and quantify dim levels of protein expression that characterize cells under the EndMT transition. Lineage tracing experiments are a preferable approach, although the specificity of the driver represents also a caveat (*Kovacic et al., 2019*; *Li et al., 2018*; *Piera-Velazquez and Jimenez, 2019*).

Our results using flow cytometry, a technique that allowed us to finely track shifts in the expression of markers associated with EndMT, showed changes in ECs, 'transitioning cells', and MyoFBs that demonstrate a role of Mφ-derived MT1-MMP in TGFβ1-mediated EndMT after MI. This conclusion is supported by the identification of ERG$^+$SMA$^+$ cells with active SMAD2 signaling within the infarct and by the results with in vitro experiments showing that addition of neutralizing anti-TGFβ1 antibody to EC-Mφ co-cultures abolished EndMT. Our findings are in line with the role of MT1-MMP as an activator of epithelial to mesenchymal transition (EMT) in other pathophysiological contexts such as development, cancer, and lung fibrosis (*Garmon et al., 2018*; *Nguyen et al., 2016*; *Xiong et al., 2017*). Interestingly, a recent scRNAseq study carried out on healthy and post-infarcted hearts identified a subset of post-MI Mφs with a HIF1α-dependent signature, which specifically upregulated *Mmp14* expression (*Dick et al., 2019*). The *Mmp14* expression in these cells converges with a hypoxia-driven program, which represents a well-known stimulus for EndMT (*Evrard et al., 2016*), further supporting the implication of *Mmp14*-expressing post-MI Mφs in EndMT. Moreover, Mφs have recently been reported as inducers of EndMT in atherosclerosis (*Helmke et al., 2019*).

Based on our discoveries, we propose that MT1-MMP mediated TGFβ1-activation in Mφs after ischemic injury contributes, among other possible TGFβ-regulated processes, to local EndMT and the generation of MyoFBs during granulation tissue formation. These cells might then take part in fibrosis, giving rise to a dense fibrotic scar that compromises LV cardiac function after MI. Restraining Mφ-mediated TGFβ1 activation by MT1-MMP thus limits EndMT, favoring preservation of cardiac microvasculature, improving myocardial blood flow, and reducing tissue hypoxia and fibrotic scarring (*Figure 7*). These alterations in MAC-Mmp14 KO mice limited LV dilation and dysfunction and suggest novel approaches to the promotion of cardiac recovery after MI. Therapeutic strategies would consist of manipulating Mφ MT1-MMP production in order to control EndMT and likely other profibrotic TFGβ effects, thus promoting angiogenesis and moderating scar formation. Interestingly, in

an experimental rat model of MI, treatment with menstrual-blood–derived mesenchymal stem cells has been shown to protect endothelial function, reduce infarct size, decrease cardiac fibrosis, and downregulate TGFβ1/SMAD signaling, all through the comprehensive inhibition of EndMT (*Zhang et al., 2013*). Following this rationale, previous studies have pointed to the beneficial effect of pharmacological inhibition of MMPs on post-MI cardiac remodeling; however, the implementation of clinically relevant therapies has proved difficult (*Creemers et al., 2001*; *Hudson et al., 2006*; *Yabluchanskiy et al., 2013*). A recent study reported that selective MT1-MMP inhibition rescued tissue damage and mortality in influenza-infected mice, demonstrating the potential of specific MT1-MMP inhibitors to ameliorate the detrimental effects of this protease on tissue remodeling (*Talmi-Frank et al., 2016*). Controlling dysregulated myocardial MT1-MMP activity in Mφs could be a suitable option for patients at risk of developing heart failure after MI.

# Materials and methods

## Key resources table

| Reagent type (species) or resource | Designation | Source or reference | Identifiers | Additional information |
|---|---|---|---|---|
| Strain, strain background (*Mus musculus*) | C57BL/6 | Charles Rivers | | |
| Genetic reagent (*Mus musculus*) | *Mmp14*<sup>f/f</sup> | *Gutiérrez-Fernández et al., 2015* | MGI:5694577 | Dr. Carlos López-Otín |
| Genetic reagent (*Mus musculus*) | Lyz2-Cre | *Clausen et al., 1999* | MGI:1934631 | |
| Genetic reagent (*Mus musculus*) | Cdh5-Cre<sup>ERT2</sup> | *Sörensen et al., 2009* | MGI:3848982 | |
| Genetic reagent (*Mus musculus*) | R26TdTomato | *Madisen et al., 2010* | MGI:3809524 | |
| Cell line (*Homo sapiens*) | HEK293 | Sigma | | |
| Antibody | Anti-CD16/CD32 (rat monoclonal) | BD Biosciences | 553141 | (1:100) |
| Antibody | Anti-CD45 (rat monoclonal) | Biolegend | 103132 | (1:100) |
| Antibody | Anti-CD45 (rat monoclonal) | Biolegend | 103116 | (1:100) |
| Antibody | Anti-CD45 (rat monoclonal) | eBioscience | 48–0451 | (1:100) |
| Antibody | Anti-CD11b (rat monoclonal) | BD Biosciences | 552850 | (1:100) |
| Antibody | Anti-CD11b (rat monoclonal) | BD Biosciences | 557395 | (1:100) |
| Antibody | Anti-CD11b (rat monoclonal) | Biolegend | 101206 | (1:100) |
| Antibody | Anti-Ly6C (rat monoclonal) | BD Biosciences | 560595 | (2:100) |
| Antibody | Anti-Ly6C (rat monoclonal) | BD Biosciences | 553104 | (1:100) |
| Antibody | Anti-CD31 (rat monoclonal) | BD Biosciences | 551262 | (1:100) |
| Antibody | Anti-CD31 (rat monoclonal) | BD Biosciences | 553372 | (1:100) |
| Antibody | Anti-PDGFR-β (rat monoclonal) | BioLegend | 136005 | (2:100) |
| Antibody | Anti-PDGFR-β (rat monoclonal) | BioLegend | 136007 | (2:100) |
| Antibody | Anti-Feeder Cells (MEFSK4) (rat monoclonal) | Miltenyi | 130-120-166 | (2:100) |
| Antibody | Anti-Feeder Cells (MEFSK4) (rat monoclonal) | Miltenyi | 130-120-802 | (2:100) |
| Antibody | Anti-F4/80 (rat monoclonal) | BioLegend | 123114 | (3:100) |
| Antibody | Anti-F4/80 (rat monoclonal) | BioLegend | 123110 | (3:100) |
| Antibody | Anti-Phospho-Smad2 (rabbit polyclonal) | Cell Signaling | 3104 | (1:100) |

*Continued on next page*

*Continued*

| Reagent type (species) or resource | Designation | Source or reference | Identifiers | Additional information |
|---|---|---|---|---|
| Antibody | Streptavidin-Alexa 488 conjugate | ThermoFisher Scientific | S11223 | (1:500) |
| Antibody | Anti-LAP (mouse monoclonal) | Biolegend | 141405 | (3:100) |
| Antibody | Anti-TGFβ1 (rabbit polyclonal) | Abcam | ab92486 | (1:100) |
| Antibody | anti-rabbit IgG (chicken polyclonal) | ThermoFisher | A-21441 | (1:500) |
| Antibody | anti- rabbit IgG (goat polyclonal) | ThermoFisher | A-21245 | (1:500) |
| Antibody | anti-CAI-IX (rabbit polyclonal) | Abcam | ab15086 | (1:100) |
| Antibody | anti-rabbit IgG (goat polyclonal) | ThermoFisher | A-11035 | (1:500) |
| Antibody | anti-CD31 (rat monoclonal) | Dianova | DIA-310 | (1:200) |
| Antibody | anti-Rat IgG (goat polyclonal) | ThermoFisher | A-11006 | (1:500) |
| Antibody | anti-pSMAD2 (rabbit monoclonal) | Cell Signaling | 3108 | (1:100) |
| Antibody | anti-SMA (mouse monoclonal) | Sigma-Aldrich | C6198 | (1:400) |
| Antibody | anti-ERG (rabbit monoclonal) | Abcam | ab196149 | (1:100) |
| Antibody | Anti-PDGFRβ (rat monoclonal) | Thermofisher | 14-1402-82 | (1:100) |
| Antibody | Anti-rat IgG (goat polyclonal) | Thermofisher | A-11006 | (1:500) |
| Antibody | Anti-TGFβ1 (rabbit polyclonal) | Santa Cruz | sc-146 | (1:500) |
| Antibody | Anti-TfR (rabbit polyclonal) | Abcam | ab84036 | (1:1000) |
| Antibody | Anti-α-tubulin (mouse monoclonal) | Sigma | T6074 | (1:1000) |
| Antibody | Anti-TGFβ1 (mouse monoclonal) | inVivoMab/Bio X Cell | BE0057 | 100 μg/mL |
| Antibody | anti-ICAM2 (rat monoclonal) | BD Biosciences | 553325 | (1:500) |
| Antibody | anti-rabbit IgG (goat polyclonal) | Jackson | 111-035-003 | (1:7500) |
| Antibody | anti-mouse IgG (goat polyclona) | Jackson | 115-035-003 | (1:7500) |
| Recombinant DNA reagent | p3TP-lux | *Wrana et al., 1992* | RRID:Addgene_11767 | Dr. Carmelo Bernabeu |
| Sequence-based reagent | qPCR primers | This paper | | *Supplement file 1* |
| Peptide, recombinant protein | Human TFGβ1 | Peprotech | 100–21 | |
| Peptide, recombinant protein | IL4 | Peprotech | 214–14 | |
| Peptide, recombinant protein | Murine M-CSF | Peprotech | 315–02 | |
| Chemical compound, drug | LPS | Sigma | L2654 | |
| Chemical compound, drug | Tamoxifen | Sigma | T5648 | |
| Chemical compound, drug | Collagenase type IV | Sigma | C5138 | |
| Chemical compound, drug | Collagenase type I | Worthington | LS004194 | |

*Continued on next page*

*Continued*

| Reagent type (species) or resource | Designation | Source or reference | Identifiers | Additional information |
|---|---|---|---|---|
| Chemical compound, drug | Collagenase type II | Worthington | LS004174 | |
| Chemical compound, drug | Porcine pancreatin | Sigma | P3292 | |
| Chemical compound, drug | RBC Lysis buffer solution | eBioscience | 00-4333-57 | |
| Chemical compound, drug | Passive Lysis 5× Buffer | Promega | E1941 | |
| Chemical compound, drug | TRIzol Reagent | Thermofisher | 15596026 | |
| Chemical compound, drug | Fluoromont-G | Southern Biotech | 0100–01 | |
| Commercial assay or kit | Dynabeads Sheep Anti-Rat IgG | Thermofisher | 11035 | |
| Commercial assay, kit | Foxp3/Transcription Factor Staining Buffer Set | eBioscience | 00-5523-00 | |
| Commercial assay, kit | High Capacity cDNA Reverse Transcription Kit | Applied Biosystems | 4368814 | |
| Commercial assay, kit | Bradford Assay | Bio-Rad | 5000001 | |
| Commercial assay, kit | Luciferase Assay System | Promega | PR-E1500 | |
| Software, algorithm | Vevo 2100 | Visual Sonics | RRID:SCR_015816 | |
| Software, algorithm | Fiji | fiji.sc | RRID:SCR_002285 | |
| Software, algorithm | GraphPad Prism | www.graphpad.com | RRID:SCR_002798 | |
| Software, algorithm | FlowJo | www.flowjo.com | RRID:SCR_008520 | |
| Software, algorithm | qBASE+ (Biogazelle) | www.qbaseplus.com | RRID:SCR_003370 | |
| Software, algorithm | NDP.view2 | Hamamatsu Photonics | | |
| Software, algorithm | Zen2 | Zeiss | RRID:SCR_013672 | |
| Software, algorithm | 3D fully automated image analysis | *Gkontra et al., 2018* | | |

## Mice

All the animals used in this study were on the C57BL/6 background. Experiments were performed in 8- to 12-week-old both male and female mice, unless otherwise indicated, kept in a specific pathogen-free (SPF) facility at Centro Nacional de Investigaciones Cardiovasculares (CNIC) under a 12 hr light/dark cycle (lights on from 07:00 to 19:00 hr), with water and chow available ad libitum. All animal procedures were conducted in accordance with EU Directive 86/609/EEC and approved by the Animal Subjects Committee of the Instituto de Salud Carlos III (Madrid, Spain) and Madrid Community Organs in the PROEX 188/26. Animals used in this study were C57BL/6 mice (Charles River), *Lyz2*-Cre mice (*Clausen et al., 1999*), and *Mmp14*^f/f mice (*Gutiérrez-Fernández et al., 2015*). *Lyz2*-Cre⁺/*Mmp14*^f/f mice (MAC-Mmp14 KO) lack *Mmp14* in Mφs; while *Lyz2*-Cre⁻/*Mmp14*^f/f littermates were used as WT controls. For endothelial lineage tracing experiments, we used *Cdh5*-Cre^ERT2 mice (*Sörensen et al., 2009*) crossed with R26TdTomato mice (*Madisen et al., 2010*). Tamoxifen (Sigma-Aldrich) was dissolved in corn oil (15 mg / mL) and injected intraperitoneally (0.15 mg tamoxifen/g mouse body weight) 5 days before LAD-ligation surgery.

## Mouse models of MI

### Left anterior descending coronary artery ligation (LAD-ligation)

Permanent ligation of the left anterior descendent (LAD) coronary artery was performed as previously described (*Kolk et al., 2009*). Briefly, mice were anesthetized with sevoflurane (5% for induction, 2–3% for maintenance), and intubated using a 24-gauge intravenous catheter with a blunt end. Mice were artificially ventilated with a mixture of $O_2$ and air (1:1 [vol/vol]) using a rodent ventilator (minivent 845) with 160 strokes/min in a total volume of 250 µL. The mouse was placed on heating pad to maintain body temperature at 37°C. A thoracotomy was performed through the fourth left intercostal space, the pericardium was opened, and the heart was exposed. The proximal LAD coronary artery was permanently ligated with a 7/0 silk suture (Lorca Marín). The thorax and the skin

incision were closed with 6/0 silk sutures (Lorca Marín) and buprenorphine (0.01 mg/kg, Buprex, Merck) was given for pain relief. Mice were sacrificed by $CO_2$ inhalation 3, 7, or 28 days post-MI. Animals not subjected to surgery were included as the physiological condition (day 0).

### Ischemia-reperfusion (I/R)

I/R protocol was performed as previously described (*Inserte et al., 2019*), following the same anesthetic and thoracotomy protocol as in permanent LAD-ligation. Once the heart was exposed, the LAD coronary artery was ligated approximately 1 mm below the edge of the left atrial appendage with an 8/0 silk suture (EthiconEndo-surgery, OH, USA). Regional ischemia was verified by visual pallor and QRS alterations within the first seconds of occlusion. After occlusion for 45 min, the suture was loosened to start reperfusion. The thorax and the skin incision were closed with 6/0 silk sutures (Lab Arago, Spain) and buprenorphine (0.01 mg/kg, Buprex, Merck) was given for pain relief. Mice with lack of ST-elevation during ischemia or lack of ST-recovery at reperfusion were excluded from further evaluation. Mice were sacrificed by $CO_2$ inhalation 21 days post-MI. Animals not subjected to surgery were included as the physiological condition (day 0).

For the two models of MI, mice with less than two affected LV segments after the surgery in terms of contractility were considered non-properly infarcted and excluded from the study.

## Echocardiographic analysis

Transthoracic echocardiography was performed on mice subjected to LAD-ligation or I/R at basal, 1, 7, and 21 or 28 days post-surgery. Infarct size was monitored by echocardiography 1 day after surgery to allow comparison of infarcts between genotypes. Shaved mice were anesthetized by inhalation of isoflurane and oxygen (1.25% and 98.75%, respectively) and placed in a biofeedback warming station that maintained core body temperature. Anesthesia depth was adjusted to maintain heart rate between 450 and 550 beats per minute. Warm ultrasound gel was applied to the chest of the animals, and echocardiography measurements were obtained using the VEVO 2100 high frequency ultrasound system with a linear transducer MS400 18–38 MHz (Visual Sonics, Toronto, Canada). Parasternal long- and short-axis views at three levels (base, middle, and apex) in two-dimensional and M-mode were obtained as described previously (*Cruz-Adalia et al., 2010*). The LV end-systolic and end-diastolic volume (LVVols and LVVold, respectively) were acquired from the parasternal two-dimensional long-axis view, and LV ejection fraction (LVEF) was calculated using the area-length method (*Ram et al., 2011*). Wall thickness at the end of the systole and diastole was measured from the M-mode short-axis view. The analysis was performed off-line by two blinded echocardiography experts.

Regional kinetic abnormalities within the LV were assessed. LV wall motion score index (WMSI) was calculated to assess global and regional cardiac function by a 12-based segment model, considering parasternal two-dimensional short and long-axis views at the three levels, as previously described (*Gallego-Colon et al., 2016*). In each level, the LV was further divided into four segments (anterior, lateral, posterior, and septal) and every segment was scored according to its severity in terms of contraction as 1 (normal), 2 (hypokinesia), 3 (akinesia), 4 (dyskinesia or aneurysm). Infarct size was estimated as the percentage of individual segments scored >1 (reflecting contractility abnormalities) over the total of LV segments, and WMSI was defined as the ratio of the sum of the score of every segment over the total number of segments evaluated.

## Flow cytometry and cell sorting

Cardiac single-cell suspensions were obtained as previously described (*Alonso-Herranz et al., 2019*). Briefly, mice were euthanized by $CO_2$ fixation and immediately perfused by intracardiac injection of cold PBS. Right and left atria were removed and the whole ventricles were minced with fine scissors and digested in collagenase IV 0.1% (528 U/mg Sigma) in PBS at 37°C for 45 min under gentle shaking. Cells were then filtered through nylon mesh of 100 μm (BD biosciences) to obtain a homogeneous cell suspension and were subjected to red blood cell lysis with RBC Lysis buffer solution (eBioscience). Single-cell suspensions were Fc-blocked using anti-mouse CD16/CD32 antibody (BD Pharmingen) 10 min at 4°C in FACS buffer (PBS 2% FBS 5 mM EDTA). Antibodies were incubated for 30 min at 4°C in FACS buffer. Where appropriate, cells were further incubated with streptavidin conjugates for 30 min at 4°C. For nuclear pSMAD2 staining, cells were fixed and

permeabilized using a commercial kit (Foxp3/Transcription Factor Staining Buffer Set, eBioscience). Flow cytometry studies were performed in a BD FACSCantoTM II flow cytometer (BD BioSciences) and analyzed with FlowJo Software (Tree Star). Cell sorting was performed with BD FACS-ARIATM II cell sorter (BD Biosciences).

After pre-selection in side scatter (SSC) *versus* forward scatter (FSC) dot plot to exclude debris and doublets, cardiac Mφs were identified as CD45$^+$CD11b$^+$F4/80$^+$ cells, ECs as CD45$^-$CD31$^+$-PDGFRβ$^-$ cells, 'transitioning' cells as CD45$^-$CD31$^{low}$PDGFRβ$^{low}$ cells, and VSMCs and MyoFBs as CD45$^-$CD31$^-$PDGFRβ$^+$ cells. Within the CD45$^+$CD11b$^+$ cells, the F4/80$^+$Ly6C$^{low}$ cardiac Mφs were sorted at 0, 7, and 28 days post-MI, whereas F4/80$^+$Ly6C$^{high}$ cardiac Mφs were purified for 3 days post-MI.

Fluorescence minus one (FMO) controls were included during acquisition for gating analyses to distinguish positive from negative staining cell populations. The standardized median fluorescence intensity (MFI) of TGFβ1, LAP, and pSMAD2 for each cardiac cell type was calculated as previously described (*Maecker et al., 2004*):

$$\text{Standardized MFI} = \frac{(\text{median}_{\text{positive}} - \text{median}_{\text{FMO}})}{2 * \text{SD}_{\text{FMO}}}$$

Whereas *median$_{positive}$* is the median intensity of the positive cell population, *median$_{FMO}$* is the median intensity of the FMO, and *SD$_{FMO}$* is the standard deviation of the intensity of the FMO.

## RNA isolation and quantitative real-time PCR (qPCR)

Cells were lysed with TRIzol Reagent (Ambion) for RNA isolation. Total RNA was isolated from at least three independent biological replicates, and RNA quality and quantity measured using the NanoDrop ND100 (Thermo Scientific). Total RNA was reverse-transcribed to cDNA using the High Capacity cDNA Reverse Transcription Kit (Applied Biosystems). qPCR analysis was performed using Sybr Green probes in the AB7900 FAST 384 Detection System (Applied Biosystems), according to the manufacturer´s instructions. Gene expression values were normalized to the housekeeping genes *36b4* and *Cyclophilin*, and expressed as relative mRNA level. Data were analyzed by qBASE+ software (Biogazelle) obtaining the Ct of the amplification products. Primer sequences are provided in *Supplementary file 1*.

## Histology and immunohistochemistry

For histological analysis, hearts were perfused with cold PBS, fixed in 4% PFA overnight, and embedded in paraffin. Transverse sections (5 µm) were stained with hematoxylin-eosin (H and E) and Masson trichrome stain according to standard procedures.

For immune-labeling of arterioles, samples were stained with anti-SMA, afterward with appropriate HRP conjugated antibody, and finally revealed with 3,3'-diaminobenzidine (DAB) following standard protocols. Whole slide images were acquired with a digital slide scanner (Hamamatsu, Nanozoomer-RS C110730) and then visualized, and exported to TIFF images using NDP.view2 software (Hamamatsu Photonics).

Manual and automated quantifications were performed with Fiji Image J Software (NIH, https://imagej.nih.gov/ij/). The infarct zone (IZ), and the remote zone (RZ) were defined on the basis of H and E-stained sections. In particular, areas containing dying or dead CMs (picnotic or absent nuclei, wavy fibers) or fibrotic areas were defined as IZ, whilst the RZ was considered the healthy LV free wall. Measurements were performed on sections obtained from the midpoint of the infarct.

## Multi-Photon microscopy and second harmonic generation imaging

Collagen fibers were visualized in H and E-stained heart sections with a Zeiss LSM 780 microscope coupled to a Spectra-Physics Mai Tai DS [pulse <70 ps] laser, by second harmonic generation (SHG) and multi-photon excitation fluorescence (MPEF) microscopy imaging technique (*Abraham et al., 2010*). Optical sections were acquired every 3 µm (25× objective) and stitched using Zen 2 software (Zeiss). The images were then stacked and flattened with Image J software to create maximum intensity Z-projections. Collagen density was calculated using a simple threshold pixel counting method within the IZ and expressed as the area fraction (% SHG). Skewness (asymmetry of pixel distribution)

and kurtosis (gray-tone spread-out distribution) were assessed as indicative of fiber arrangement (*Mostaço-Guidolin et al., 2013*).

## Tissue immunofluorescence and confocal microscopy

Transverse paraffin sections (7 µm) of infarcted hearts were deparaffinized, rehydrated, and finally washed in PBS 5 min twice. Antigen retrieval was performed by means of pH = 6 citrate buffer for 20 min in the microwave at maximum intensity. Afterward, sections were cooled down at room temperature (RT) for 1 hr 30 min and then washed with PBS 5 min twice. Sections were blocked for 1 hr at RT (0.3% Triton X-100, 5% goat serum, and 5% BSA in PBS) and primary antibodies were incubated overnight at 4° C (0.3% Triton X-100, 2.5% goat serum, and 2.5% BSA in PBS). Next, sections were washed with 0.1% Triton X-100 in PBS at RT for 10 min three times and secondary antibodies and DAPI (1/5000) for nuclear staining were then incubated for 1 hr 30 min at RT (0.3% Triton X-100, 2.5% goat serum, and 2.5% BSA in PBS). After four washing steps with 0.1% Triton X-100 in PBS at RT for 10 min plus 10 min more with PBS alone, slides were mounted with Fluoromont-G (Southern Biotech). Images were acquired with a Nikon A1R confocal microscope with sections every 1.5 µm. Three to four areas were acquired within the IZ (the two most distal edges and the center of the infarcted area), and the other three to four within the RZ (LV free wall most distal to the infarct).

For the in vivo EndMT identification, transverse paraffin thick sections (14 µm) of 7 day-post-infarcted hearts were sequentially stained. For sequential immunostaining, primary rabbit pSMAD2 was incubated overnight at 4°C (0.3% Triton X-100, 2.5% goat serum, and 2.5% BSA in PBS). After washing, goat anti-rabbit Alexa 488 was incubated 1 hr at RT (0.3% Triton X-100, 2.5% goat serum, and 2.5% BSA in PBS). Sections were afterward thoroughly washed, and additional blocking was performed with 0.3% Triton X-100, 5% rabbit serum, and 5% BSA in PBS 1 hr at RT. Directly labeled primary antibodies (mouse anti-SMA-Cy3 and rabbit anti-ERG-647) and DAPI were then incubated 1 hr at RT. Finally, sections were washed and mounted as indicated before. Visual co-localization of the three markers (ERG, SMA, and pSMAD2,) was performed with the orthogonal view plug-in of ImageJ.

For CA-IX quantifications, the total area covered by the CA-IX signal in every image was normalized by the total area, and this ratio was averaged in between individuals. For pSMAD2 quantification in VSMCs and MyoFBs, DAPI nuclei within arteriolar SMA signal were considered VSMCs and DAPI nuclei in non-arteriolar SMA signal were considered MyoFBs. The number of $DAPI^+pSMAD2^+$ VSMCs and the number of $DAPI^+pSMAD2^+$ MyoFBs were normalized by the total number of VSMCs or MyoFBs, respectively, and expressed as a percentage. Finally, the percentages obtained in each region were averaged within individuals. Measurements were performed on sections obtained from the midpoint of the infarct.

## 3D fully automated microvasculature image analysis

Transverse paraffin sections (15 µm) of infarcted hearts were stained with anti-CD31 and anti-SMA antibodies and DAPI for nuclear staining. Two to four images were acquired within the IZ (including the two most distal edges and the center of the IZ) per mouse with a Nikon A1R confocal microscope. Thus, 38 3D images were quantified. Characterization of the microvasculature was performed by means of a fully automated pipeline (*Gkontra et al., 2018*). Modules of the pipeline were adapted as previously described (*Żak et al., 2019*) in order to account for the different animal model (mouse) as well as the use of the relatively thin tissue sections compared to the thick tissue sections of ~100 µm used in the original work. Moreover, since images included both tissue areas and areas belonging to the microscope slide, it was necessary to separate the tissue volume from the background glass area. For this purpose, first the images were denoised by means of non-local means filtering (*Buades et al., 2005*). Subsequently, 3D tissue segmentation was performed by applying the Otsu multi-level thresholding technique (*Otsu, 1979*) on the nuclei channel. It should be noted that the intensity levels used for thresholding varied between three and four depending on the image and they were automatically defined. Voxels of all intensities levels but the lower one were considered to belong to the tissue. Finally, holes within the segmented area were automatically identified and filled. It should be noted that apart from normalization purposes, the tissue mask was used to exclude from the corresponding segmentation mask of vessels and $SMA^+$ cells non-tissue voxels that were incorrectly identified as belonging to vessels or SMA, respectively. To this end,

element-wise multiplications of the segmentation of vessels and SMA$^+$ cells with the tissue mask were performed.

Following tissue segmentation, we applied modules of the pipeline that permit the automatic segmentation of nuclei, vessels, and SMA$^+$ cells from the corresponding image channels. Classification of microvessels according to their size and relation with SMA$^+$ cells into different physiologically meaningful categories was performed as previously described (Żak et al., 2019). Finally, quantitative parameters regarding the morphology and angioarchitecture of the network as well as the relation of vessels with SMA$^+$ cells were extracted. The parameters are summarized in *Table 3*.

## Cell culture

Bone marrow-derived Mφs (BMDMs) were harvested from 8 to 10 week-old WT and MAC-Mmp14 KO mice by flushing femurs and tibias with PBS. Cells were filtered through a 100 µm nylon cell strainer (Falcon) and cultured in RPMI (Lonza) supplemented with 20% L929-cell conditioned medium, 10% FBS (Gibco), 1% penicillin and streptomycin (P/S, Lonza), and 1 mM L-Gln (Lonza) in sterile non-tissue culture treated 10 cm Petri dish. BMDMs were grown to confluence and activated overnight with 50 ng/mL LPS (Sigma) to stimulate TGFβ1 production. The medium was then replaced, and after incubation for 48 hr, the supernatant was harvested, centrifuged, and stored at −20°C until use in co-culture experiments. BMDMs were polarized towards the M1 or M2 phenotype by incubation for 24 hr with 10 ng/mL LPS (Sigma) or 10 ng/mL IL4 (Peprotech), respectively.

Mouse aortic endothelial cells (MAECs): Six to eight aortas from 4-week-old male C57BL/6 mice were pooled to obtain a single-cell suspension (Fogelstrand et al., 2009). Briefly, after carefully aorta dissection and fat removal under a microscope, aortas were incubated in collagenase type I solution (3.33 mg/mL, Worthington) for 5 min at 37°C. Then, adventitia was removed with forceps, the aortas were then cut into small pieces (1–2 mm) and incubated for 45 min at 37°C in a collagenase type I (6 mg/mL, Worthington) and elastase (2.5 mg/mL, Worthington) solution. The obtained single-cell suspension was plated on 0.5%-gelatin-coated plates in DMEM/F12 supplemented with 20% FBS, 2 mM L-Gln, 1% P/S, and ECGS/H. When culture became confluent, MAECs were positively selected with an antibody against intercellular adhesion molecule 2 (anti-ICAM2, BD Biosciences) coupled to magnetic beads (Dynabeads Sheep Anti-Rat IgG, ThermoFisher), and cultured as indicated in M199 supplemented with 20% FBS, 1% P/S, 2 mM L-Gln 1% HEPES, and ECGS/H (Koziol et al., 2012).

Cardiac FBs were isolated from 8-week-old C57BL/6 mice as previously described (Brand et al., 2010). Briefly, hearts were sequentially incubated in a solution containing collagenase type II (0.48 mg/mL, Worthington) and porcine pancreatin (2.5 mg/mL, Sigma) at 37°C in a water bath with gentle shaking. Once tissue was completely disaggregated, cell fractions were resuspended in DMEM supplemented with 10% FBS, 1% P/S, and 2 mM L-Gln. The cell suspension was then plated on 1%-gelatin-coated plates and incubated at 37°C for 1 hr to allow FBs to attach. The supernatant, enriched in the myocyte fraction, was then removed and replaced with a fresh medium.

Neonatal CMs were isolated from P1-P3 C57BL/6 pups as previously described (Brand et al., 2010). Briefly, 25–30 neonates were sacrificed, and hearts were digested in a solution containing collagenase type II (0.48 mg/mL, Worthington) and porcine pancreatin (2.5 mg/mL, Sigma) at 37°C in a water bath with gentle shaking. Once the tissue was completely segregated, cell fractions were resuspended in DMEM supplemented with 10% M199 medium, 15% FBS, 0.5% HEPES, 1% P/S, and 2 mM L-Gln. The cell suspension was then plated on 1%-gelatin-coated plates and incubated at 37°C for 1 hr to remove FBs through attachment to the plate. The supernatant, enriched in the myocyte fraction, was then centrifuged and, resuspended in the same medium, and CMs were plated on 1%-gelatin–coated plates. CM beating was observed in the cultures.

Human embryonic kidney 293 (HEK293) cells (Lonza) were cultured in DMEM (Lonza) with 10% FBS.

Cardiac Mφs: FACS-sorted Mφs (CD45$^+$CD11b$^+$F4/80$^+$Ly6C$^{low}$ cells) were purified from WT and MAC-Mmp14 KO hearts on day 7 after MI. They were cultured in DMEM 10% FBS 10 ng/mL M-CSF (Peprotech).

## Cell extract preparation and western blot analysis

WT or MAC-Mmp14 KO BMDMs were treated with 1 ng/mL LPS (Sigma) for 24 hr to stimulate TGFβ1 production. Prior to lysate collection, cell viability was determined by microscopy under bright field and cell debris removed by two washes of monolayers with PBS. For preparation of total lysates, $4 \times 10^6$ BMDMs were incubated in RIPA buffer, containing 50 mM Tris-HCl, pH 8; 150 mM NaCl; 1% Triton X-100; 0.5% sodium deoxycholate; 0.1% SDS; 1 mM PMSF (Sigma), and a protease and phosphatase inhibitor cocktail (Sigma), for 30 min at 4° C in the rotor. The lysate was centrifuged (14,000 g, 10 min), and the supernatant containing the proteins was transferred to a new tube. For subcellular fractioning, $4 \times 10^6$ BMDMs were incubated in HES buffer, containing 20 mM HEPES, 1 mM EDTA, 250 mM sucrose, and a protease and phosphatase inhibitor cocktail (Sigma), for 10 min at 4° C. Cells were lysed through a 22G needle followed by a 25G needle, and centrifuged at 500 g for 8 min to pellet the unbroken cells. Then, the supernatant was transferred to an empty tube and centrifuged at 10,000 g for 12 min to separate the cytosolic (supernatant) and the membrane (pellet) protein fractions.

Protein concentration was estimated using Bradford Assay (Bio-Rad) and 30 µg of total protein were separated by 10% SDS-PAGE and transferred to a nitrocellulose membrane (Bio-Rad). Correct protein loading was confirmed by Ponceau staining. Membranes were incubated overnight with anti-bodies against TGFβ1 (Santa Cruz), TfR (Abcam), and α-tubulin (Sigma), and then thoroughly washed and incubated with HRP-conjugated anti-rabbit (Jackson) or anti-mouse (Jackson) antibodies (1:7500). Blots were visualized using the chemiluminescent Immobilon Classico Western HRP sub-strate (Millipore). Chemiluminescent signal was detected using ImageQuant LAS 4000 and densitometry analysis performed using ImageJ. The ratio between proteins of interest (TGFβ1) and endogenous control (TfR) was calculated for data normalization. Graphs represent fold change calculated based on normalized data.

## HEK293 transfection and luciferase assay

HEK293 cells (40%–50% confluence) were transfected over 6 hr with the plasmid p3TP-lux (*Wrana et al., 1992*) with 2.5M CaCl$_2$ in HEPES buffered saline (HBS). This plasmid contains luciferase downstream of the PAI-1 promoter, therefore the luciferase activity is proportional to the production of bioactive TGFβ1 (*Abe et al., 1994*). Luciferase assay was carried out as described (*Abe et al., 1994*). In brief, transfected HEK293 cells were plated at a density of 30,000 cells per well in p96-well plates and allowed to adhere for 6 hr. Then, 10 ng/mL of recombinant TGFβ1 (Peprotech) for the positive control, 40,000 WT BMDMs or 40,000 MAC-Mmp14 KO BMDMs (previously activated with 50 ng/mL LPS overnight) were added to the plate. After 24 hr of incubation, cells were lysed with passive lysis buffer (Promega), and luciferase activity was assayed with the Promega Luciferase Assay according to manufacturer's instructions.

When the assay was carried out with cardiac Mφs, CD45⁺CD11b⁺F4/80⁺Ly6Cˡᵒʷ cells were sorted as indicated in *Figure 1—figure supplement 1A-B* from 7 days post-MI WT or MAC-Mmp14 KO hearts. 50,000 cardiac Mφs were added to the previously transfected and plated HEKs (as indicated above), and M-CSF (10 ng/mL) was kept in the medium for the 24 hr of culture to maintain cardiac Mφs alive. Then, cells were lysed with passive lysis buffer (Promega), and luciferase activity was assayed with the Promega Luciferase Assay according to manufacturer's instructions.

## Lentiviral transduction

The full-length MT1-MMP sequence (FL) or mutated version to disable catalytic activity (E240A) was cloned into the SFFV-IRESGFP lentiviral backbone. Lentiviruses expressing Mock, MT1-MMP FL, or MT1-MMP E240A were prepared and titered as previously described (*Esteban et al., 2020*). For viral inoculation, we incubated MAC-Mmp14 KO BMDMs with the viral supernatants (MOI = 10) in RPMI supplemented with 20% L929-cell-conditioned medium, 10% FBS, 1% P/S, and 1 mM L-Gln for 48 hr. GFP signal was detected by fluorescent microscopy in transduced BMDMs. We then removed the viral supernatants and stimulated the cells with 50 ng/mL LPS overnight. Afterward, the media was removed and cells were cultured in DMEM 10% FBS for 24 hr to obtain conditioned media. Conditioned media was then added to HEK293 cells for luciferase assay as described above.

## Co-cultures and qPCR assay

FBs were plated at 500,000 per well on 1%-gelatin-coated p12-well plates in DMEM supplemented with 10% FBS, 1% P/S, and 2 mM L-Gln, and allowed to attach for 2 hr. The medium was then removed and FBs were treated with WT or MAC-Mmp14 KO BMDM supernatants for 72 hr. Afterward, cells were lysed with TRIzol Reagent (Ambion) for RNA isolation.

CMs were plated at 200,000 per well on 1%-gelatin-coated p12-well plates in DMEM supplemented with 10% M199 medium, 15% FBS, 0.5% HEPES, 1% P/S, and 2 mM L-Gln. Once CM beating was observed (after 4 days of culture), cells were treated with WT or MAC-Mmp14 KO BMDM supernatants for 72 hr. Afterward, cells were lysed with TRIzol Reagent (Ambion) for RNA isolation.

MAECs were plated at 150,000 per well on 0.5%-gelatin–coated optical p6-well plates in M199 supplemented with 20% FBS, 1% P/S, 2 mM L-Gln, 1% HEPES, and ECGS/H, and allowed to attach for 6 hr. The medium was then removed and treatments were added in the same medium with reduced serum (5% FBS). The MAEC cultures were overlaid with 250,000 LPS-activated WT or MAC-Mmp14 KO BMDMs. Co-cultures were maintained for 4 days. As a positive control for EndMT, MAEC monocultures were treated with 10 ng/mL TGFβ1. Afterward, cells were lysed with TRIzol Reagent (Ambion) for RNA isolation following manufacturer's instructions.

## Co-culture and immunofluorescence assay

MAECs were plated at 20,000 per well on 0.5%-gelatin–coated optical p96-well plates in M199 supplemented with 20% FBS, 1% P/S, 2 mM L-Gln, 1% HEPES, and ECGS/H, and allowed to attach for 6 hr. The medium was then removed and treatments were added in the same medium with reduced serum (5% FBS). The MAEC cultures were overlaid with 30,000 LPS-activated WT or MAC-Mmp14 KO BMDMs, or with 50,000 cardiac Mφs from 7 days post-MI WT or MAC-Mmp14 KO mice. MAECs were also treated with TGFβ1 (10 ng/mL) and/or neutralizing αTGFβ1 antibody (100 μg/mL). Co-cultures were maintained for 4 days.

After two washes in HBBS, cultures were fixed for 10 min at room temperature (RT) in 4% paraformaldehyde. Cells were then permeabilized for 10 min at RT in 0.2% Triton X-100 in phosphate-buffered saline (PBS) and blocked for 1 hr at RT in 5% goat serum and 5% BSA in PBS. After overnight incubation with primary antibodies at 4° C in 2.5% goat serum and 2.5% BSA in PBS, cells were washed three times with PBS at RT and then incubated with secondary antibodies and DAPI for 1 hr 30 min at RT in 2.5% goat serum and 2.5% BSA in PBS. Cells were then washed four times with PBS. Cells were visualized under a Nikon A1R microscope, and images were acquired as $2 \times 2$ tile scans of $5 \times 1$ μm sections. Maximum intensity projections were used for quantifications.

## Statistics

Data are presented as mean ± SEM. Unpaired *t*-test was used when two groups were compared, and a comparison of more than two sets of data was done using analysis of variance (ANOVA) with Tukey's post-test. All statistical analyses were performed using Prism v7 (GraphPad Software, La Jolla, CA). Differences were considered significant when $p < 0.05$, and represented as *$p < 0.05$, **$p < 0.01$, ***$p < 0.001$, and ****$p < 0.0001$.

## Acknowledgements

We thank Ángel Colmenar, Carmen Contreras, and the CNIC Cellomics, Proteomics, Microscopy, Advanced Imaging, and Animal Units for technical support and Simon Bartlett for English editing. We also thank the Proteomics Unit at the Vall d'Hebron Institute of Oncology. We thank Carlos López-Otín (Universidad de Oviedo) for providing the MT1-MMP1[f/f] mice, Ralf Adams (Max Planck Institute of Molecular Biomedicine) for providing the *Cdh5*-Cre[ERT2] mice, and Carmelo Bernabeu (Centro de Investigaciones Biológicas, CIB-CSIC) for the p3TP-lux plasmid. This study was supported by grants from the Spanish Ministry of Science, Innovation and Universities (SAF2017-90604-REDT-NurCaMeIn, RTI2018-095928-BI00 to MR; SAF2017-83229-R to AGA), Comunidad de Madrid (MOIR-B2017/BMD-3684) to MR, and La Marató de TV3 Foundation to DGD, AGA, and MR. LAH is funded by a fellowship from La Caixa-CNIC. ASE is supported by La Residencia de Estudiantes and funded by a fellowship from La Caixa and a partnership between FORD-España and Apadrina La Ciencia. Ana Paredes is funded by a fellowship from the Ministry of Science and Innovation (BES-

2016-076632). The CNIC is supported by the MCNU and the Pro CNIC Foundation and is a Severo Ochoa Center of Excellence (SEV-2015–0505).

## Additional information

### Funding

| Funder | Grant reference number | Author |
| --- | --- | --- |
| Spanish Ministry of Science, Innovation and Universities | SAF2017-90604-REDT-NurCaMeIn | Mercedes Ricote |
| Spanish Ministry of Science, Innovation and Universities | RTI2018-095928-BI00 | Mercedes Ricote |
| Spanish Ministry of Science, Innovation and Universities | SAF2017-83229-R | Alicia G Arroyo |
| Comunidad de Madrid | MOIR-B2017/BMD-3684 | Mercedes Ricote |
| La Marato TV3 Foundation | | David García-Dorado Alicia G Arroyo Mercedes Ricote |
| Fundacion La Caixa | | Laura Alonso-Herranz Álvaro Sahún-Español |
| La Residencia de Estudiantes | | Álvaro Sahún-Español |
| FORD-Spain and Apadrina La Ciencia | | Álvaro Sahún-Español |
| Spanish Ministry of Science and Innovation | BES-2016-076632 | Ana Paredes |

The funders had no role in study design, data collection and interpretation, or the decision to submit the work for publication.

### Author contributions

Laura Alonso-Herranz, Conceptualization, Data curation, Software, Formal analysis, Validation, Investigation, Visualization, Methodology, Writing - original draft; Álvaro Sahún-Español, Pilar Gonzalo, Data curation, Software, Formal analysis, Investigation, Visualization, Methodology, Writing - review and editing; Ana Paredes, Marta Cedenilla, Javier Inserte, Formal analysis, Investigation, Methodology; Polyxeni Gkontra, Data curation, Software, Formal analysis, Methodology, Writing - review and editing; Vanessa Núñez, Cristina Clemente, Methodology; María Villalba-Orero, Software, Formal analysis, Validation, Visualization, Methodology; David García-Dorado, Conceptualization, Resources, Investigation, Writing - review and editing; Alicia G Arroyo, Conceptualization, Resources, Data curation, Formal analysis, Supervision, Funding acquisition, Validation, Investigation, Writing - original draft, Project administration, Writing - review and editing; Mercedes Ricote, Conceptualization, Resources, Supervision, Funding acquisition, Validation, Investigation, Writing - original draft, Project administration, Writing - review and editing

### Author ORCIDs

Laura Alonso-Herranz https://orcid.org/0000-0003-0880-4735
Álvaro Sahún-Español https://orcid.org/0000-0003-3833-5386
Ana Paredes https://orcid.org/0000-0003-3119-8788
Pilar Gonzalo https://orcid.org/0000-0001-8811-8369
Cristina Clemente https://orcid.org/0000-0002-5831-9132
Alicia G Arroyo https://orcid.org/0000-0002-1536-3846
Mercedes Ricote http://orcid.org/0000-0002-8090-8902

### Ethics

Animal experimentation: All animal procedures were conducted in accordance with EU Directive 86/609/EEC and approved by the Animal Subjects Committee of the Instituto de Salud Carlos III

(Madrid, Spain) and Madrid Community Organs in the PROEX 188/26. All surgery was performed under anesthesia with sevoflurane (5% for induction, 2%-3% for maintenance) and buprenorphine (0.01 mg/kg, Buprex, Merck & Co. Inc) was given for pain relief.

## Decision letter and Author response
Decision letter https://doi.org/10.7554/eLife.57920.sa1
Author response https://doi.org/10.7554/eLife.57920.sa2

## Additional files
### Supplementary files
- Supplementary file 1. List of primer sequences used in the study.
- Transparent reporting form

### Data availability
All data generated or analysed during this study are included in the manuscript and supporting files. Source data files have been provided for all the figures.

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
