## [Decision Letter]

**Acceptance summary:**

The novel concept reported in this manuscript is that macrophage-specific deletion of membrane type 1 matrix metalloprotease (MT1-MMP) will limit the possible adverse effects induced by excessive collagenolytic activities by MMPs. This manuscript provides mechanistic insights how MT1-MMP degrades the latency associated peptide (LAP) allowing TGF-β to act on endothelial cells, promoting EndMT and contributing to adverse tissue remodeling, suggesting macrophage MT1-MMP holds potential as a therapeutic target for cardiac ischemic disease.

**Decision letter after peer review:**

Thank you for submitting your article "Macrophages promote endothelial-to-mesenchymal transition via MT1-MMP/TGFβ1 after myocardial infarction" for consideration by *eLife*. Your article has been reviewed by two peer reviewers, including Noriaki Emoto as the Guest Reviewing Editor and Reviewer #1, and the evaluation has been overseen by a Senior Editor.

The reviewers and the Guest Reviewing Editor have discussed the reviews and found the study of considerable interest. The main points have been summarized below to help you prepare a revised submission.

Summary:

The manuscript "Macrophages promote endothelial-to-mesenchymal transition via MT1-MMP/TGFβ1 after myocardial infarction" is a study that highlights the role of specific macrophage knockdown of MT1-MMP in cardiac repair and remodeling after myocardial infarction. This manuscript is well written and systematic. By using reliable methods, the authors found interesting data that support their conclusions.

Cardiac fibrosis after myocardial infarction is a needed process to prevent the rupturing of ventricle wall, however, excessive extracellular matrix (ECM) remodeling leads to altered chamber compliance, thereby compromising cardiac output. On the other hand, over collagenolytic activities by MMPs, such as MT1-MMP, result in deleterious effects in animal models (Koenig, et al., 2012). The novel concept that the authors propose is macrophages specific deletion of MT1-MMP will limit the adverse effects.

Here, the authors provide the mechanisms that are distinct from MT1-MMP's collagenolytic activity, they prove that MT1-MMP degrade the latency associated peptide (LAP), so that the TGFβ acts on endothelial cells, promoting EndMT and contributing to adverse tissue remodeling.

The study is based on extensive data and overall is well performed and interesting.

Essential revisions:

1) The major weakness of the manuscript is that the authors failed to prove the causal contribution of EndMT inhibition to the improved cardiac function and reduction of fibrosis in MT1-MMP mutant mice. While a change in smooth muscle marker gene expression is shown in ECs at day 7, no long term data are provided and the transition of EndMT to fibroblasts is not shown. Since monocytes and specifically TGFß have multiple functions in tissue repair, it is plausible that additional other pathophysiological processes are regulated by MT1-MMP. To solve this question will require lineage tracing of the mutant mice, which takes a while. Thus, if the authors cannot provide these data they should adapt the conclusion of the manuscript and the Abstract. The possible contribution of other effector pathways need to be highlighted. Question to be addressed or discussed may include an effect of MT1-MMP on p16INK4a and p21CIP1/WAF1 and a premature aging phenotype or a more detailed analysis of M1 versus M2 macrophages. The manuscript would profit from an open minded analysis of the phenotype.

2) One major technical weakness is the use of PDGFRß, which preferentially marks pericytes. Thus, the population of EndMT positive cells determined likely represents a subset of EndMT-undergoing cells. PDGFRa is a more global fibroblast marker and could have been used for sorting of fibroblast-like ECs. This limitation needs to be discussed and a panel of additional myofibroblast markers should be used to verify the data. Histological examination is essential. Antibodies can be false positive or negative and demonstration of appropriate localization makes the data stronger and it needs to be assessed if pericytes are specifically affected by the MT-MMP1 lacking fibroblasts.

3) It would be very interesting to see whether the MT1-MMP in macrophages could have any additional effects in other adjacent cells in the ventricle, other than the stated TGFβ-mediated EndMT that affected the microvascular ECs, that can cause the reported amelioration in adverse cardiac remodeling seen in Figure 1. Since it is known that macrophages are recruited to the infarcted site, the reviewer wonders whether any other remodeling-related effects caused by factors secreted by macrophage with altered expression of MT1-MMP could also happen in the cardiomyocytes or cardiac fibroblast cells. For example, using conditioned medium derived from macrophages that overexpress or with silenced MT1-MMP and treating it to cardiomyocytes or cardiac fibroblasts in vitro to see any remodeling-related changes in the cells could help explain whether the phenomenon caused by MT1-MMP macrophages in cardiac remodeling was strictly happening due to the changes in ECs or not.

Specific comments:

1) Figure 1D: please provide raw data (2-deltaδ CT) allowing the comparison of the MT1-MMP level in the different cell types.

2) The authors should show that anti-TGFß blocks EndMT in vitro to confirm the proposed mechanism of action.

3) The raw data should also be included in the time course studies measuring cardiac function. How was infarct size after operation controlled?

---

## [Author Response]

Essential revisions:1) The major weakness of the manuscript is that the authors failed to prove the causal contribution of EndMT inhibition to the improved cardiac function and reduction of fibrosis in MT1-MMP mutant mice. While a change in smooth muscle marker gene expression is shown in ECs at day 7, no long term data are provided and the transition of EndMT to fibroblasts is not shown. Since monocytes and specifically TGFß have multiple functions in tissue repair, it is plausible that additional other pathophysiological processes are regulated by MT1-MMP. To solve this question will require lineage tracing of the mutant mice, which takes a while. Thus, if the authors cannot provide these data they should adapt the conclusion of the manuscript and the Abstract. The possible contribution of other effector pathways need to be highlighted. Question to be addressed or discussed may include an effect of MT1-MMP on p16INK4a and p21CIP1/WAF1 and a premature aging phenotype or a more detailed analysis of M1 versus M2 macrophages. The manuscript would profit from an open minded analysis of the phenotype.

Although lineage tracing experiments would serve to support the contribution of Mφ MT1MMP to EndMT in vivo, crossing an EC-reporter mouse (e.g. Cdh5CreERT-Rosa26TdTomato) with our Lyz2-Cre-Mmp14^f/f^ mouse to carry out these experiments would be time-consuming and would not substantially alter the take-home message. Taking into account the editor’s recommendations, we have performed the experiments suggested by the reviewers and have adjusted the Abstract and conclusions of the paper to reflect the new data and to provide a more open-minded interpretation of the phenotype (Abstract, Introduction and Discussion). We discuss the possibility that reduced TGFβ1 activation by Mφ MT1-MMP could result in attenuation of FB activation to MyoFBs (subsection “The inactivation of Mφ MT1-MMP reduces TGFβ1-pSMAD2 signaling in cardiac ECs, MyoFBs, and VSMCs after MI” and Discussion, third paragraph) and loss of TGFB1-mediated angiostatic effects (Discussion, third paragraph), ultimately contributing to the ameliorated cardiac dysfunction reported in the KO mice.

As the reviewers suggested, we have explored other possible mechanisms contributing to the improved cardiac phenotype of MAC-Mmp14 KO mice. We have performed a detailed analysis of pro-inflammatory (M1) and anti-inflammatory (M2) phenotypes in WT and KO Mφs, finding no differences in Mφ polarization between WT and MAC-Mmp14 KO mice. This result supports the idea that *Mmp14* deletion does not affect the Mφ M1/M2 activation spectrum (Figure 1—figure supplement 4). These new data are included in the subsection “*Mmp14* inactivation in Mφs does not influence their phenotype” and Figure 1—figure supplement 4 of the revised manuscript.

Moreover, following the reviewers’ recommendations, we have explored whether deletion of *Mmp14* may have an effect on p16^INK4a^ and p21^CIP1^ (alternatively p21^WAF1^)-induced senescence in Mφs, as previously suggested in cardiomyocytes, fibroblasts, and adipose tissue (Gutiérrez-Fernández et al., 2015). We studied the expression of *Cdkn1a* (p21) and *Cdkn2a* (p16) and a panel of senescence-associated secretory phenotype (SASP)-related genes, again finding no differences between genotypes (Author response image 1), suggesting no effects of Mφ MT1-MMP deletion on Mφ senescence induction. *Cdkn2a* (p16) was not expressed either in 7-day-post-MI cardiac Mφs or in BMDMs. Given the controversy about the exclusivity of p16 activation for senescence in Mφs (Hall et al., 2017; Liu et al., 2019) and the lack of differences between WT and MAC-Mmp14 KO mice in SASP-related genes, we think that including these data would distract from the focus of the manuscript and would not contribute to understanding of the phenotype.

**Author response image 1. sa2fig1:** (A) mRNA expression levels of genes related to senescence-associated secretory phenotype (SASP) by qPCR in FACS-sorted 7-day-post-MI cardiac Mφs (CD45^+^CD11b^+^F4/80^+^Ly6C^low^ cells) from WT and MAC-Mmp14 KO mice. Data are means ± SEM of 4 mice per genotype. Unpaired t-test. (B) mRNA expression levels of genes related to SASP by qPCR in BMDMs from WT and MAC-Mmp14 KO mice. Data are means ± SEM of 4 mice per genotype. Unpaired t-test.

2) One major technical weakness is the use of PDGFRß, which preferentially marks pericytes. Thus, the population of EndMT positive cells determined likely represents a subset of EndMT-undergoing cells. PDGFRa is a more global fibroblast marker and could have been used for sorting of fibroblast-like ECs. This limitation needs to be discussed and a panel of additional myofibroblast markers should be used to verify the data. Histological examination is essential. Antibodies can be false positive or negative and demonstration of appropriate localization makes the data stronger and it needs to be assessed if pericytes are specifically affected by the MT-MMP1 lacking fibroblasts.

We understand the reviewers’ concern, and have now included new data and discussed the literature to provide more robust evidence for the appropriateness of PDGFRβ as a marker of MyoFBs (subsection “The inactivation of Mφ MT1-MMP reduces TGFβ1-pSMAD2 signaling in cardiac ECs, MyoFBs, and VSMCs after MI” and Figure 4—figure supplement 1). First, we selected PDGFRβ as MyoFB marker because Dr. Alicia G. Arroyo’s lab has previously demonstrated perfect correlation between SMA and PDGFRβ in porcine infarcted myocardium and clear pericyte and MyoFB histological identification with PDGFRβ (Gkontra et al., 2018). Other studies have also demonstrated that PDGFRβ expression is an early feature of MyoFB activation (Henderson et al., 2013). To further validate our findings on EndMT based on PDGFRβ, we have complemented our quantification of EndMT cells (CD31^+^PDGFRβ^+^ cells) using the MEFSK4 antibody (Figure 5—figure supplement 3).

This antibody labels almost all PDGFRα^+^, Col1α1^+^ cardiac fibroblasts in flow cytometry analysis, although it also labels a small subpopulation of pericytes (Pinto et al., 2016). In addition, our qPCR analysis confirmed PDGFRβ^+^ cell expression of the MyoFB markers *Tagln, Col1a1, Col2a1*, and *Col31a* (Figure 5—figure supplement 1).

We agree with the reviewer on the importance of histological examination of PDGFRβ+ cells in the myocardium. We have now performed co-staining of PDGFRβ, SMA, CD31, and DAPI. In line with our previous papers (Gkontra et al., 2018; Żak et al., 2019), this analysis showed that PDGFβ indeed labels two distinct cell subsets in the infarcted myocardium: i) PDGFβ+ cells around vessels (pericytes and VSMCs); and ii) non-perivascular PDGFβ+ cells (MyoFBs) (Figure 4—figure supplement 1). We could not undertake a histological examination of MEFSK4+ cells due to the lack of valid MEFKS4 antibodies for immunohistochemistry (Humeres and Frangogiannis, 2019).

3) It would be very interesting to see whether the MT1-MMP in macrophages could have any additional effects in other adjacent cells in the ventricle, other than the stated TGFβ-mediated EndMT that affected the microvascular ECs, that can cause the reported amelioration in adverse cardiac remodeling seen in Figure 1. Since it is known that macrophages are recruited to the infarcted site, the reviewer wonders whether any other remodeling-related effects caused by factors secreted by macrophage with altered expression of MT1-MMP could also happen in the cardiomyocytes or cardiac fibroblast cells. For example, using conditioned medium derived from macrophages that overexpress or with silenced MT1-MMP and treating it to cardiomyocytes or cardiac fibroblasts in vitro to see any remodeling-related changes in the cells could help explain whether the phenomenon caused by MT1-MMP macrophages in cardiac remodeling was strictly happening due to the changes in ECs or not.

We agree that we cannot exclude a possible contribution of effects on other adjacent cell types to the amelioration of cardiac dysfunction in MAC-Mmp14 KO mice. We appreciate the editor’s suggestion, and to address this point, we have cultured fibroblasts (Figure 4—figure supplement 2A) and cardiomyocytes (Figure 4—figure supplement 2B) with supernatants from WT and MACMmp14 KO BMDMs. We found no significant differences in fibrosis, apoptosis, fibroblast activation, or cardiomyocyte-related markers between stimulation with conditioned medium from WT or KO Mφs. Thus, no major changes in fibroblasts or cardiomyocytes result from crosstalk with Mφs that lack MT1-MMP. These data are included in the revised version of the manuscript (subsection “The inactivation of Mφ MT1-MMP reduces TGFβ1-pSMAD2 signaling in cardiac ECs, MyoFBs, and VSMCs after MI” and Figure 4—figure supplement 2) and discussed in the third paragraph of the Discussion.

Specific comments:1) Figure 1D: please provide raw data (2-sΔ CT) allowing the comparison of the MT1-MMP level in the different cell types.

We have now included the required raw data (2^-∆∆Ct) for calculation of *Mmp14* expression levels in the different cell types (Figure 1—figure supplement 2D, and Figure 1—figure supplement 2—source data 7).

2) The authors should show that anti-TGFß blocks EndMT in vitro to confirm the proposed mechanism of action.

We are extremely grateful to the editor for the suggestion to block EndMT in vitro with neutralizing anti-TGFβ1. The revised version of the manuscript includes two new panels in Figure 6—figure supplement 1C-D showing the result of treating MAEC-BMDM co-cultures with neutralizing anti-TGFβ1 antibody to confirm the proposed TGFβ1-mediated mechanism. Treatment of MAEC-BMDM co-culture with anti-TGFβ1 neutralizing antibody blocked EndMT transition, demonstrating that Mφs induce EndMT via MT1-MMP/TGFβ1. We also blocked TGFβ1-induced EndMT with anti-TGFβ1 antibody to prove the efficiency of the blocking antibody (subsection “Mφ-derived MT1-MMP induces EndMT after MI”, Discussion, fifth paragraph and Figure 6—figure supplement 1C-D).

3) The raw data should also be included in the time course studies measuring cardiac function. How was infarct size after operation controlled?

The raw data from the echocardiography analysis are provided in the Figure 1—source data 1. Infarct size after operation was monitored by echocardiography (1 day after surgery) to allow comparison of infarcts between genotypes and this is now explicitly stated in the manuscript (subsection “Echocardiographic analysis”). These data are now included in the Figure 1—source data 1 excel file.

References:

Hall, B.M., Balan, V., Gleiberman, A.S., Strom, E., Krasnov, P., Virtuoso, L.P., Rydkina, E., Vujcic, S., Balan, K., Gitlin, II, et al. (2017). p16(Ink4a) and senescence-associated β-galactosidase can be induced in macrophages as part of a reversible response to physiological stimuli. Aging 9, 1867-1884.

Humeres, C., and Frangogiannis, N.G. (2019). Fibroblasts in the Infarcted, Remodeling, and Failing Heart. JACC Basic Transl Sci *4*, 449-467.

Liu, J.-Y., Souroullas, G.P., Diekman, B.O., Krishnamurthy, J., Hall, B.M., Sorrentino, J.A., Parker, J.S., Sessions, G.A., Gudkov, A.V., and Sharpless, N.E. (2019). Cells exhibiting strong p16 INK4a promoter activation in vivo display features of senescence. 116, 2603-2611.